# Noise-Adaptive Layerwise Learning Rates: Accelerating Geometry-Aware Optimization for Deep Neural Network Training

## Abstract

Geometry-aware optimization algorithms, such as Muon, have achieved remarkable success in training deep neural networks (DNNs). These methods leverage the underlying geometry of DNNs by selecting appropriate norms for different layers and updating parameters via norm-constrained linear minimization oracles (LMOs). However, even within a group of layers associated with the same norm, the local curvature can be heterogeneous across layers and vary dynamically over the course of training. For example, recent work shows that sharpness varies substantially across transformer layers and throughout training, yet standard geometry-aware optimizers impose fixed learning rates to layers within the same group, which may be inefficient for DNN training.

In this paper, we introduce a *noise-adaptive layerwise learning rate* scheme on top of geometry-aware optimization algorithms and substantially accelerate DNN training compared to methods that use fixed learning rates within each group. Our method estimates gradient variance in the dual norm induced by the chosen LMO *on the fly*, and uses it to assign time-varying noise-adaptive layerwise learning rates within each group. We provide a theoretical analysis showing that our algorithm achieves a sharp convergence rate. Empirical results on transformer architectures such as LLaMA and GPT demonstrate that our approach achieves faster convergence than state-of-the-art optimizers.

## 1 Introduction

Optimization algorithms are cornerstones for modern deep learning, enabling the training of increasingly large neural networks, such as LLaMA (Touvron et al., 2023) and GPT (Achiam et al., 2023) models. While standard optimizers such as SGD (Robbins & Monro, 1951) and Adam (Kingma & Ba, 2014) remain widely used, they often overlook the geometry of neural network parameter spaces. Recently, geometry-aware optimization algorithms such as Muon (Jordan et al., 2024) have demonstrated remarkable empirical success by performing orthogonalized updates on matrix parameters. Building on this idea, Pethick et al. (2025) developed a framework that selects appropriate norms for different layers and updates parameters via norm-constrained linear minimization oracles (LMOs). These methods go beyond standard optimizers by exploiting structural properties (e.g. layer-wise operator norms) of DNNs rather than treating all parameters uniformly, thus leading to improved performance and acceleration for large-scale foundation model pretraining (Liu et al., 2025).

Despite their success, most of the existing geometry-aware optimizers simply assign fixed learning rates within groups of layers associated with the same norm choice. However, these algorithms neglect the heterogeneous and dynamic nature of various layers during the neural network training. For example, recent studies (Wang et al., 2025) have shown that sharpness or local curvature of the objective function can vary substantially across different types of layers (e.g., query-key (QK) layers, value-output (VO) layers, and multilayer perceptron (MLP) in transformers). Moreover, these variations evolve over time, as observed when training with AdamW (Loshchilov & Hutter, 2017). (Riabinin et al., 2025) firstly proposed layerwise learning rates for the geometry-aware optimization methods based on smoothness parameters. In contrast, we focused on the noise magnitude of each layer instead of the smoothness parameters. In particular, we have observed similar phenomena in training a LLaMA model with the Muon optimizer[1]. Figure 1 highlights that the stochastic gradient

---

[1] We follow `https://github.com/KellerJordan/modded-nanogpt` to apply Muon optimizer to the transformer hidden layers (including query, key, value, output, MLP layers), and AdamW to the embedding, LM head, normalization layers.

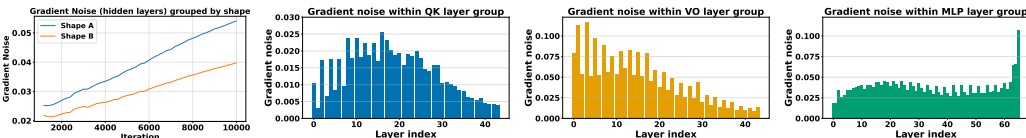

Figure 1: The stochastic gradient noise is heterogeneous across groups and layers in transformers. The first subfigure shows that average gradient noise in hidden layers varies across parameter groups defined by matrix shape and evolves over training. The last three subfigures illustrate that, within each layer group, the gradient noise varies substantially across layers[3].

noise differs substantially across layer groups or layers, and shifts throughout training. Nevertheless, state-of-the-art geometry-aware optimizers such as D-Muon (Liu et al., 2025) and Scion (Pethick et al., 2025) use the same fixed learning rate for matrices of the same shape, ignoring the fact that gradient noise on layers with the same shape can vary significantly over iterations as shown in Figure 1. This mismatch suggests that treating such layers uniformly may lead to inefficient training, motivating the need for novel layerwise learning rate schemes.

Layerwise adaptive learning rates (You et al., 2017; 2019) are widely used in deep learning under standard Euclidean spaces. These optimizers automatically rescale updates according to gradient magnitudes, which reduces manual tuning and often accelerates convergence. However, they disregard the structural geometry of neural networks by treating all parameters as if they belonged to the same category. In reality, neural networks contain diverse parameter groups such as matrices in attention layers, vectors in bias terms, and embedding tables, where different layers in each group exhibit vastly different noise profiles as illustrated in our Figure 1. The key open question is how to design adaptive learning rates beyond standard Euclidean spaces, enabling geometry-aware optimizers to exploit heterogeneous gradient noise across layers and over the course of training.

In this paper, we propose a new geometry-aware optimization algorithm named *Lanton: LAyer-wise Noise-adaptive learning raTe scaling with Operator Norms*. Our algorithm dynamically estimates gradient variance in the dual norm induced by the chosen LMO and uses this estimate to assign layerwise learning rates that adapt over the course of training. Unlike existing approaches, which treat all layers in a group uniformly, our algorithm accounts for the heterogeneity of gradient noise across layers, leading to smaller learning rates for layers with larger gradient noise, thereby enabling finer-grained and more efficient optimization. Importantly, the proposed mechanism is compatible with the geometry-aware optimizers, such as Muon (Jordan et al., 2024) and D-Muon (Liu et al., 2025). Our contribution can be summarized as follows.

- We propose a new optimization algorithm named *LANTON: LAyer-wise Noise-adaptive learning raTe scaling with Operator Norms*, which can dynamically capture the gradient noise of each layer and thus accordingly rescale the learning rate of each layer.

- We prove that our method achieves a sharp convergence rate of $\tilde{O}(1/\sqrt{T}+\sqrt{\sum_\ell \bar{\sigma}_\ell}/T^{1/4})$, where $\bar{\sigma}_\ell$ denotes an upper bound on the gradient noise of the layer $\ell$. Our bound shows improved noise dependence under the layer-wise noise assumption. By explicitly accounting for the heterogeneous noise levels across layers, our analysis demonstrates the advantage of noise-adaptive layer-wise learning rates.

- Empirically, we evaluate our approach on language model training and image classification, including LLaMA, GPT2 and convolutional neural network, and show that it substantially accelerates training and improves sample efficiency compared to state-of-the-art optimizers. Our results indicate that dynamically adapting learning rates at the layer level can better capture the evolving optimization landscape, leading to faster convergence and improved training efficiency. Together, these contributions highlight the importance of integrating noise adaptivity into geometry-aware optimization and open new directions for scalable and effective training of deep neural networks.

## 2 RELATED WORK

A long line of work has studied optimization for deep learning. The most classical method is SGD (Robbins & Monro, 1951). Early advances focused on adaptive learning rates, including Adagrad

---

[3]See Appendix F for the implementation details.

(Duchi et al., 2011), RMSProp (Tieleman & Hinton, 2012), Adadelta (Zeiler, 2012), and the widely used Adam (Kingma & Ba, 2014). Later developments improved Adam in various ways: AdamW (Loshchilov & Hutter, 2017) introduced decoupled weight decay and has become the default choice for deep learning; several variants incorporate variance reduction, such as AdEMAMix (Pagliardini et al., 2024) and MARS-AdamW (Yuan et al., 2024); others target memory efficiency, including Adafactor (Shazeer & Stern, 2018), Lion (Chen et al., 2023), MeZO (Malladi et al., 2023), GaLore (Zhao et al., 2024a), Adam-mini (Zhang et al., 2024), and Signum (Zhao et al., 2024b).

Another line of work approximates or leverages second-order information. K-FAC (Martens & Grosse, 2015) and Shampoo (Gupta et al., 2018) are classical examples. The substantial compute and memory overheads of second-order optimizers have motivated distributed implementations of Shampoo (Anil et al., 2020; Shi et al., 2023). More recently, lightweight preconditioned optimizers such as Sophia (Liu et al., 2023a) and SOAP (Vyas et al., 2024) have been proposed, achieving substantial speedups over AdamW in large-scale language model pretraining.

A third research direction focuses on layer-wise or block-wise learning rates to accelerate training. LARS (You et al., 2017) and LAMB (You et al., 2019) are widely used for large-batch training, while more recent approaches extend AdamW with blockwise learning rates (Wang et al., 2025).

Several parameter-free or schedule-free optimizers aim to reduce the burden of hyperparameter tuning, including Dog (Ivgi et al., 2023), Prodigy (Mishchenko & Defazio, 2023), and Schedule-Free AdamW (Defazio et al., 2024).

Most recently, the theory of modular duality in optimization and the perspective of steepest descent under different operator norms (Bernstein & Newhouse, 2024a;b; Large et al., 2024) have inspired the design of matrix-based and geometry-aware optimizers, including Muon (Jordan et al., 2024) and Scion (Pethick et al., 2025), as well as distributed implementations such as D-Muon (Liu et al., 2025) and Dion (Ahn et al., 2025), which further improve training efficiency and stability at scale.

## 3 PRELIMINARIES

In this work, we consider the stochastic optimization problem $\min_X f(X) := \mathbb{E}_{\xi \in \mathcal{D}}[F(X; \xi)]$, where $\xi$ is random noise sampled from an unknown distribution $\mathcal{D}$, and $X \in \mathcal{S}$ is the model parameter, where $X = [X_1, \ldots, X_p]$, $X_i \in \mathcal{S}_i := \mathbb{R}^{m_i \times n_i}$, and $\mathcal{S} := \bigotimes_{i=1}^{p} \mathcal{S}_i$. Similarly, write the gradient as $\nabla f(X) = [\nabla_1 f(X), \ldots, \nabla_p f(X)] \in \mathcal{S}$, and the stochastic gradient as $\nabla F(X; \xi) = [\nabla_1 F(X; \xi), \ldots, \nabla_p F(X; \xi)] \in \mathcal{S}$ (here we adopt the notation and setup from (Riabinin et al., 2025). We assume that the objective is bounded from below, i.e., $f^* := \inf_X f(X) > -\infty$.

**Notations.** Let $\| \cdot \|$ denote an arbitrary (not necessarily Euclidean) vector/matrix norm with associated dual norm $\| \cdot \|_*$, and let $\| \cdot \|_{\text{nuc}}$ denote the nuclear norm. We use $\langle \cdot, \cdot \rangle$ for the trace inner product, defined as $\langle A, B \rangle = \text{tr}(A^\top B)$ for $A, B \in \mathbb{R}^{m \times n}$. For two positive functions $f$ and $g$, we write $f \lesssim g$ (resp. $f \gtrsim g$) if there exists $c > 0$ such that $f(x) \leq cg(x)$ (resp. $f(x) \geq cg(x)$) for all $x$. We use standard big-O notation, with $\tilde{O}$ and $\tilde{\Omega}$ used to hide polylogarithmic factors, respectively.

**Linear Minimization Oracle (LMO).** The LMO is a fundamental concept in convex optimization (Frank et al., 1956), particularly in the context of algorithms like the Frank-Wolfe algorithm (also known as the conditional gradient method (Jaggi, 2013)). Given a convex feasible set $\mathcal{K}$ and a direction vector/matrix $u$, the LMO returns an extreme point of $\mathcal{K}$ that minimizes the linear function $\langle u, x \rangle$ over $\mathcal{K}$. Mathematically, this can be expressed as: $\text{LMO}(u) = \arg\min_{x \in \mathcal{K}} \langle u, x \rangle$.

Throughout this paper, we focus on the special case where $\mathcal{K} := \{x \mid \|x\| \leq 1\}$ for some chosen (not necessarily Euclidean) norm $\| \cdot \|$ (Pethick et al., 2025), unless specified otherwise.

**Operator Norm and RMS Norm.** Given a matrix $A \in \mathbb{R}^{m \times n}$ and two normed vector spaces $(\mathbb{R}^n, \| \cdot \|_a)$ and $(\mathbb{R}^m, \| \cdot \|_b)$, the "$a$ to $b$" induced operator norm is defined as $\|A\|_{a \to b} := \max_{x \in \mathbb{R}^n, x \neq 0} \frac{\|Ax\|_b}{\|x\|_a} = \sup_{\|x\|_a = 1} \|Ax\|_b$. Given a vector $x \in \mathbb{R}^d$, the RMS norm is defined as $\|x\|_{\text{RMS}} := \frac{1}{\sqrt{d}} \|x\|_2$.

## 4 OUR METHOD

**Algorithmic Framework.** Our proposed algorithmic framework (Algorithm 1) consists of three main stages at each iteration. First (lines 4-6), we compute the stochastic gradient $G_t^\ell$ for each layer,

**Algorithm 1** LANTON: LAyer-wise Noise-adaptive raTe scaling with Operator Norms

1: **Input:** $X_0, \alpha, \beta_1, \beta_2, \gamma, \eta, G_0 = \nabla F(X_0; \xi_0), B_0 = G_0$
2: **while** $t < T$ **do**
3:    **for** each layer $\ell$ **do**
4:       $G_t^\ell = \nabla_\ell F(X_t; \xi_t), \tilde{G}_t^\ell = \nabla_\ell F(X_t; \tilde{\xi}_t)$          ($\tilde{G}_t^\ell$ is used only in Option II)
5:       $B_t^\ell = \beta_1 B_{t-1}^\ell + (1 - \beta_1)G_t^\ell$
6:       $O_t^\ell = \mathrm{LMO}(B_t^\ell)$      (choose norm based on $\ell$'s group $\mathcal{G}_\ell$, Table 1 line 5)
7:       $H_t^\ell = \beta_2 H_{t-1}^\ell + (1 - \beta_2) \cdot \begin{cases} \|G_t^\ell - G_{t-1}^\ell\|_*^2 & \text{Option I (practical)} \\ \|G_t^\ell - \tilde{G}_t^\ell\|_*^2 & \text{Option II (theoretical)} \end{cases}$   (Table 1 line 4)
8:       $\alpha_t^\ell = \alpha/\sqrt{\alpha^2 + H_t^\ell}, \alpha_t^m = \max_{\ell \in \mathcal{G}_\ell} \alpha_t^\ell$    (max is over $\ell$'s group $\mathcal{G}_\ell$, Table 1 line 1)
9:       $\eta_t^\ell = \eta_t \sqrt{\alpha_t^\ell/\alpha_t^m}$         ($\eta_t \in [\eta_{\min}, \eta_{\max}]$ follows a cosine decay schedule)
10:      $X_{t+1}^\ell = X_t^\ell - \eta_t^\ell O_t^\ell$
11:    **end for**
12: **end while**

Table 1: The choice of LMO can be different between layers. We use $W \in \mathbb{R}^{d_{\mathrm{out}} \times d_{\mathrm{in}}}$ to denote a matrix and $w \in \mathbb{R}^d$ to denote a vector. Write the SVD as $W = U\Sigma V^\top$.

| Parameter Group | Hidden layers (query, key, value, output, mlp) | Embedding, LM head layers | RMS norm |
|---|---|---|---|
| Size | Matrix $\in \mathbb{R}^{d_{\mathrm{out}} \times d_{\mathrm{in}}}$ | Matrix $\in \mathbb{R}^{d_{\mathrm{out}} \times d_{\mathrm{in}}}$ | Vector $\in \mathbb{R}^d$ |
| Norm $\|\cdot\|$ | RMS $\to$ RMS | $1 \to \infty$ | RMS |
| Dual Norm $\|\cdot\|_*$ | $\sqrt{d_{\mathrm{out}}/d_{\mathrm{in}}}\|\cdot\|_{\mathrm{nuc}}$ | $\|\cdot\|_{1\to 1}$ | $\sqrt{d}\|\cdot\|_2$ |
| LMO | $-\sqrt{d_{\mathrm{out}}/d_{\mathrm{in}}}UV^\top$ | $-\frac{1}{d_{\mathrm{in}}}\mathrm{sign}(W)$ | $-\sqrt{d}\frac{w}{\|w\|_2}$ |
| LMO Implementation | Newton-Schulz | Signum | RMS Normalization |

accumulate its momentum $B_t^\ell$, and then obtain the direction $O_t^\ell = \mathrm{LMO}(B_t^\ell)$ by invoking a LMO, where the choice of norm depends on the structural group of layer $\ell$ (embedding/LM head layers, hidden layers, or non-matrix layers; see Table 1). Note that line 4-6 is the same as the work of Scion (Pethick et al., 2025) and Gluon (Riabinin et al., 2025). Second (lines 7-9), the key novelty of our framework is to incorporate noise-adaptive layer-wise learning rate scaling. We maintain a momentum buffer $H_t^\ell$ to track the moving average of the estimated noise level for each layer. This buffer can be updated in two ways: a practical option (using $G_t^\ell$ and $G_{t-1}^\ell$ and avoiding extra computation) and a theoretical option (using two independent stochastic gradients $G_t^\ell$ and $\tilde{G}_t^\ell$ at each step). Based on $H_t^\ell$, the layer-wise scaling $\alpha_t^\ell$ is computed, and the effective learning rate is adjusted proportionally through the ratio $\alpha_t^\ell/\alpha_t^m$, ensuring that layers with larger noise magnitudes employ smaller learning rates. Finally (lines 10-11), we update the model parameters with the scaled stepsize and the direction given by LMO.

**Choice of Norm Constraint and LMO Implementation.** To determine appropriate norm constraints for different types of parameters in deep neural networks, we adopt the operator norm perspective recently advanced in (Large et al., 2024; Bernstein & Newhouse, 2024a; Pethick et al., 2025). As summarized in Table 1, parameters naturally fall into three groups: (i) hidden layers (e.g., query, key, value, output, and MLP weights), which are represented as matrices and we use the RMS $\to$ RMS operator norm with dual nuclear norm (scaled by $\sqrt{d_{\mathrm{out}}/d_{\mathrm{in}}}$); (ii) weight-sharing layers such as embedding and LM head matrices, where the $\ell_1 \to \ell_\infty$ operator norm is used with dual $\ell_1 \to \ell_1$ norm; and (iii) non-matrix parameters like RMS normalization vectors, where the RMS norm with dual $\ell_2$ norm (scaled by $\sqrt{d_{\mathrm{model}}}$) is adopted. These dual norms are critical in line 7 of Algorithm 1 for estimating the layer-wise gradient noise magnitude. Based on the chosen norms, the corresponding LMOs in line 6 of Algorithm 1 also differ across parameter types: for hidden layers, the LMO corresponds to a scaled $UV^\top$ computed efficiently via Newton-Schulz iterations; for embedding and LM head layers, the LMO reduces to a scaled element-wise sign operator; and for RMS normalization vectors, the LMO is implemented by RMS normalization. This unified design of norm constraints, dual norms, and LMOs with their implementations ensures both theoretical consistency with our algorithmic framework and practical efficiency in large-scale deep learning.

**Noise-Adaptive Layer-wise Learning Rates.** To capture the heterogeneous noise levels across different layers, we introduce noise-adaptive layer-wise learning rates, which dynamically scale

the stepsize of each layer according to its estimated stochastic gradient variance. Specifically, we maintain a variance tracker $H_t^\ell = \beta_2 H_{t-1}^\ell + (1 - \beta_2)\|G_t^\ell - \tilde{G}_t^\ell\|_*^2$ (line 7), where $\beta_2 \in (0,1)$ serves as a momentum-like parameter that smooths the estimate, akin to second-moment accumulation in adaptive optimizers. The resulting adaptive scaling factor $\alpha_t^\ell = \alpha/\sqrt{\alpha^2 + H_t^\ell}$ (line 8) ensures that layers subject to higher noise levels (large $H_t^\ell$) receive proportionally smaller effective learning rates, consistent with classical stochastic optimization theory. We implement this by reweighting the base learning rate with the ratio $\alpha_t^\ell/\alpha_t^m$ (where $\alpha_t^m = \max_{\ell \in \mathcal{G}_\ell} \alpha_t^\ell$), thereby aligning the updates across layers under a unified theoretical principle. While our theoretical framework (see Section 5) assumes two independent gradient estimates $G_t^\ell$ and $\tilde{G}_t^\ell$, in practice we approximate $\tilde{G}_t^\ell$ by the previous step gradient $G_{t-1}^\ell$. This avoids doubling the batch size and keeps the total number of sampled data consistent with standard baselines, thus ensuring fair comparisons in empirical evaluation.

**Comparison with Other Optimizers.** Compared to Muon (Jordan et al., 2024), Scion (Pethick et al., 2025), Gloun (Riabinin et al., 2025), and D-Muon (Liu et al., 2025), our method introduces noise-adaptive layer-wise learning rates by estimating gradient variance in the dual norm induced by the chosen LMO. Unlike Muon and D-Muon, which use AdamW for embedding and LM head layers, we adopt a geometry-aware framework (similar to Scion) and update these weight-sharing layers with Signum (see Table 1).

Optimizers such as LARS (You et al., 2017) and LAMB (You et al., 2019) also use layer-wise rescaling to stabilize large-batch training. However, these methods treat all layers uniformly. In contrast, our algorithm is geometry-aware, selecting norms tailored to hidden, embedding, and normalization layers, and updating them through LMOs with noise-adaptive scaling.

Finally, although Algorithm 1 resembles Gong et al. (2025) in estimating noise magnitude, there are key differences. Our method is LMO-based and works under arbitrary norms, while Gong et al. (2025) is restricted to the Euclidean space. Our noise adaptivity refers to per-layer scaling based on estimated variance, whereas theirs targets convergence without prior noise knowledge. Moreover, our moving-average variance estimator $H_t^\ell$ remains $O(1)$ with high probability, in contrast to their cumulative estimator $\sum_{k=1}^t \|G_k - \tilde{G}_k\|^2$ which grows as $O(\sqrt{t})$.

## 5 ANALYSIS

In this section, we provide theoretical convergence guarantees for Algorithm 1. Let $\|\cdot\|_{(\ell)}$ denote the chosen norm of layer $\ell$ with dual norm $\|\cdot\|_{(\ell)*}$, and let $p$ be the number of layers. We begin by presenting the assumption of layer-wise $L$-smoothness. Importantly, we do not assume that either the primal norm $\|\cdot\|_{(\ell)}$ or the dual norm $\|\cdot\|_{(\ell)*}$ is Euclidean. A similar layer-wise smoothness assumption is also imposed in Riabinin et al. (2025) to capture the geometry of neural networks.

**Assumption 5.1.** The objective $f$ is layer-wise $L$-smooth with constants $L := (L_1, \ldots, L_p) \in \mathbb{R}_+^p$, i.e., for all $\ell = 1, \ldots, p$, $X = [X_1, \ldots, X_p]$, and $Y = [Y_1, \ldots, Y_p]$, $\|\nabla_\ell f(X) - \nabla_\ell f(Y)\|_{(\ell)*} \leq L_\ell \|X_\ell - Y_\ell\|_{(\ell)}$.

Our second assumption states that the stochastic gradient oracle is unbiased and the layer-wise gradient noise is almost surely bounded both above and below in the dual space.

**Assumption 5.2.** (i) The stochastic gradient oracle is unbiased, i.e., $\mathbb{E}[\nabla F(X, \xi) \mid X] = \nabla f(X)$. (ii) It holds with probability one for all $\ell$ that $\underline{\sigma}_\ell \leq \|\nabla_\ell F(X, \xi) - \nabla_\ell f(X)\|_{(\ell)*} \leq \bar{\sigma}_\ell$ with $\underline{\sigma}_\ell \geq 0$.

Compared to the standard bounded variance assumption (used for expectation-based analysis) or the almost surely bounded-noise assumption (used for high-probability analysis) in stochastic optimization, Assumption 5.2 additionally requires that the stochastic gradient noise is almost surely lower bounded. A similar assumption is also made in (Gong et al., 2025). In the noisy setting, we assume $0 < \underline{\sigma}_\ell \leq \bar{\sigma}_\ell$, while in the noiseless setting we have $\bar{\sigma}_\ell = \underline{\sigma}_\ell = 0$. Note that in practice, we are always in the noisy setting where $0 < \underline{\sigma}_\ell \leq \bar{\sigma}_\ell$, as illustrated in Figure 1. From a technical perspective, this assumption is crucial for establishing a tight lower bound on $\alpha_t^\ell/\alpha_t^m$. For further proof details, see Lemma 5.5.

We now present our main result. Here $C_1, C_2$ (with $C_2 \geq 1$) are the universal constants defined in Lemma A.3, which may depend on the dimension of the model parameters. Depending on the choice of norm constraint, one may select different $C_1, C_2$ to obtain tighter dimension-dependent bounds, rather than applying a uniform choice. A detailed discussion is provided in Remark A.4.

**Theorem 5.3.** *Suppose Assumptions 5.1 and 5.2 hold. Let $\Delta_1 = f(X_1) - f^*$. Set $\beta_1 = 1 - \alpha$ with $\alpha = \min\left(\frac{\sqrt{\Delta_1 \sum_\ell L_\ell}}{\sum_\ell \bar{\sigma}_\ell \sqrt{T}}, 1\right)$, $1 - \min_\ell \frac{\sigma_\ell^4}{32(2C_2\bar{\sigma}_\ell^2 - \sigma_\ell^2)^2 \log(4T/\delta)} \leq \beta_2 < 1$, $\eta_{\max} = \sqrt{\frac{\Delta_1 \alpha}{\sum_\ell L_\ell T}}$, and $\eta_{\min} = \eta_{\max}/\kappa_\eta$ with $1 \leq \kappa_\eta \leq O(1)$. With probability at least $1 - \delta$, we have*

$$\frac{1}{T}\sum_{t=1}^{T}\sum_{\ell=1}^{p}\|\nabla_\ell f(X_t)\|_{(\ell)*} \lesssim \frac{\sqrt{C_2}(\sum_\ell \bar{\sigma}_\ell)^2}{\sqrt{\Delta_1 \sum_\ell L_\ell T}} + \frac{C_2^{3/2}}{C_1}\sqrt{\log\frac{T}{\delta}}\left(\frac{\sqrt{\Delta_1 \sum_\ell L_\ell}}{\sqrt{T}} + \frac{\sqrt{\sum_\ell \bar{\sigma}_\ell}(\Delta_1 \sum_\ell L_\ell)^{1/4}}{T^{1/4}}\right).$$

Theorem 5.3 shows that Algorithm 1 achieves a convergence rate of $\tilde{O}(1/\sqrt{T} + \sqrt{\sum_\ell \bar{\sigma}_\ell}/T^{1/4})$. Our bound highlights the advantage of adopting a layer-wise noise assumption. It achieves improved noise dependence compared to the $O(1/T^{3/4} + \sum_\ell \bar{\sigma}_{\max}/T^{1/4})^4$ bound established in (Pethick et al., 2025, Theorem 5.7), where $\bar{\sigma}_{\max}$ is the uniform noise bound assumed in prior work (Pethick et al., 2025). This improvement arises from recognizing that different layers exhibit distinct noise levels during training, and thus should not be treated uniformly. Empirically, we observe noise heterogeneity across layer groups (see Figure 1 and Table 3). Moreover, we compute that $\sqrt{\sum_\ell \bar{\sigma}_\ell} = 3.654$, which is significantly smaller than $\sum_\ell \bar{\sigma}_{\max} = 18.018$ in the LLaMA-1.1B pretraining on C4 dataset (Dodge et al., 2021), thereby validating our theoretical gain in both analysis and experiments.

## 5.1 PROOF OUTLINE

Here we give an outline of the proof of Theorem 5.3, containing the main components of our analysis; see Appendices B and C for full details. The proof sketch below is based on the setting of Theorem 5.3. To start, we introduce a few key definitions (with the convention $0/0 := 1$):

$$\kappa_\sigma^\ell = \begin{cases} \bar{\sigma}_\ell/\underline{\sigma}_\ell & \underline{\sigma}_\ell > 0 \\ 1 & \bar{\sigma}_\ell = 0 \end{cases}, \quad \kappa_\sigma = \max_\ell \kappa_\sigma^\ell, \quad \bar{\sigma}_{\max} = \max_\ell \bar{\sigma}_\ell, \quad \text{and} \quad t_0 = \frac{\log 2}{\log(1/\beta_2)}. \tag{1}$$

The following lemma provides high-probability two-sided bounds for the variance tracker $H_t^\ell$, which in turn allow us to derive tight upper and lower bounds for $\alpha_t^\ell$ (numerator of the noise ratio term). The key to the analysis is an application of the Azuma-Hoeffding inequality (see Lemma A.1).

**Lemma 5.4.** *With probability at least $1-\delta$, for all $\ell$ and $t_0 \leq t \leq T$, $\frac{\sigma_\ell^2(1-\beta_2^t)}{C_2} \leq H_t^\ell \leq 4\bar{\sigma}_\ell^2(1-\beta_2^t)$.*

With Lemma 5.4, we can effectively lower bound the noise ratio term $\alpha_t^\ell/\alpha_t^m$, which is used to assign layerwise learning rates in line 9 of Algorithm 1, with high probability. Our next lemma shows that $\alpha_t^\ell/\alpha_t^m$ is both upper and lower bounded throughout training under our assumptions. Consequently, the learning rate $\eta_t^\ell$ is bounded on both sides with high probability.

**Lemma 5.5.** *With probability at least $1 - \delta$, for all $\ell$ and $t \leq T$,*

$$\min\left\{\frac{\alpha}{\sqrt{\alpha^2 + 4\bar{\sigma}_{\max}^2}}, \frac{1}{2\sqrt{C_2}\kappa_\sigma}\right\} =: \alpha_r \leq \frac{\alpha_t^\ell}{\alpha_t^m} \leq 1, \tag{2}$$

*and therefore, with probability at least $1 - \delta$, we have $\alpha_r \eta_{\min} \leq \eta_t^\ell \leq \eta_{\max}$ for all $\ell$ and $t \leq T$.*

We now provide a high-level proof sketch of our main result. See Appendix C for full proof details.

***Proof sketch of Theorem 5.3.*** The main novelty in the proof is to leverage the magnitude of $H_t^\ell$ (Lemma 5.4) as a surrogate for the true stochastic gradient variance, ensuring that the noise-adaptive layerwise learning rate $\alpha_t^\ell$ has roughly the same magnitude as if the stochastic gradient noise were known (Lemma 5.5). The rest of the proof proceeds similarly to that of (Cutkosky & Mehta, 2020, Theorem 1) and (Li & Hong, 2025; Shen et al., 2025; Riabinin et al., 2025). Define $\hat{\epsilon}_t^\ell = B_t^\ell - \nabla_\ell f(X_t)$ and $\epsilon_t^\ell = G_t^\ell - \nabla_\ell f(X_t)$. We begin by applying Lemma 5.5 to the descent lemma (see Lemma C.1), rearranging to obtain:

$$\sum_{t=1}^{T}\sum_{\ell=1}^{p}\eta_t^\ell\|\nabla_\ell f(X_t)\|_{(\ell)*} \leq \frac{\Delta_1}{\alpha_r \eta_{\min}} + \sum_{\ell=1}^{p}\left(\frac{2\eta_{\max}}{\alpha_r \eta_{\min}}\sum_{t=1}^{T}\|\hat{\epsilon}_t^\ell\| + \frac{\eta_{\max}^2}{2\alpha_r \eta_{\min}}L_\ell T\right).$$

---

[4]This rate is obtained by replacing the global variance in (Pethick et al., 2025) with the layer-wise variance.

Using $L$-smoothness (Assumption 5.1) and standard calculations, we have

$$\|\hat{\epsilon}_{t+1}^\ell\|_{(\ell)*} \leq \beta_1^t \|\hat{\epsilon}_1^\ell\|_{(\ell)*} + (1 - \beta_1) \left\|\sum_{\tau=0}^{t-1} \beta_1^\tau \epsilon_{t-\tau}^\ell\right\|_{(\ell)*} + \eta_{\max} L_\ell \sum_{\tau=0}^{t-1} \beta_1^\tau. \tag{3}$$

Next, we apply the concentration inequality introduced in (Liu et al., 2023b, Lemma 2.4) to bound $\|\sum_{\tau=0}^{t-1} \beta_1^\tau \epsilon_{t-\tau}^\ell\|_F$, and then use the equivalence of norms (see Lemma A.3) to derive that, with probability at least $1 - \delta$,

$$\left\|\sum_{\tau=0}^{t-1} \beta_1^\tau \epsilon_{t-\tau}^\ell\right\|_{(\ell)*} \leq \frac{1}{C_1} \left\|\sum_{\tau=0}^{t-1} \beta_1^\tau \epsilon_{t-\tau}^\ell\right\|_F \leq \frac{4 C_2 \bar{\sigma}}{C_1} \sqrt{\frac{\log(2T/\delta)}{1 - \beta_1}}. \tag{4}$$

Substituting Equation (4) back into Equation (3) gives the bound for $\|\hat{\epsilon}_t^\ell\|_{(\ell)*}$. With suitable parameter choices as specified in Theorem 5.3, this concludes the proof. $\square$

## 6 EXPERIMENTS

In this section, we present the empirical results in comparison with the state-of-the-art optimizers by pretraining two mainstream transformer architectures GPT (Radford et al., 2019) and LLaMA (Touvron et al., 2023) series. The experiment of image classification is deferred to Appendix D. We include the analysis of running time in Appendix J, the ablation studies about batch size in Appendix L, the estimation method of gradient noise in Appendix M. All experiments were run on $4\times$ NVIDIA H200 graphic cards with Intel XEON Platinum 8558 CPU.

### 6.1 EXPERIMENTAL SETTINGS

**Baselines** We compare our LANTON with AdamW (Loshchilov & Hutter, 2017), Muon (Jordan et al., 2024), MARS (short for MARS-AdamW) (Yuan et al., 2024), SCION (Pethick et al., 2025), D-Muon (Liu et al., 2025), the layer-wise learning rate algorithm LAMB (You et al., 2019), and block-wise learning rate algorithm BW-AdamW (Wang et al., 2025). SCION and D-Muon apply the Muon optimizer to matrix parameters in hidden layers (e.g., query, key, value, mlp), and all these algorithms use Newton-Schulz iteration (Bernstein & Newhouse, 2024b) to approximately orthogonalize the update matrix, i.e., $UV^\top$ in Table 1.

**Models** We evaluate on both GPT and LLaMA-style decoders. For GPT we use the HuggingFace GPT2 family: GPT2-small (124M parameters) and GPT2-medium (355M parameters). For LLaMA we configure two sizes: LLaMA-0.5B and LLaMA-1.1B. Unless noted, all models are decoder-only with rotary positional embeddings and RMSNorm/LayerNorm per architecture defaults. Refer to Table 4 for detailed model configuration.

**Datasets** We pretrain GPT2 and LLaMA models on three datasets. OpenWebText-100k is used for GPT-small/medium models, and it is a subset of Openwebtext dataset (Gokaslan et al., 2019). As there is no validation set in OpenWebText-100k, we split $90\%/10\%$ into training/validation set and train models with teacher forcing. MiniPile (Kaddour, 2023) is used for LLaMA-0.5B, where minipile is a subset of the deduplicated Pile corpus (Gao et al., 2020). C4 (Colossal Clean Crawled Corpus) (Dodge et al., 2021) is a large-scale English text corpus constructed by aggressively cleaning Common Crawl webpages, and we use it to pretrain LLaMA-1.1B following the standard text-to-token pipeline. All corpora are tokenized with the model's native tokenizer.

### 6.2 TRAINING SETUP AND RESULTS

#### 6.2.1 IMPLEMENTATION OF LANTON

We implement LANTON on top of the D-Muon (Liu et al., 2025), which carefully adjusts the update magnitudes between hidden layers and non-hidden layers (embedding and LM head layers). Let $\eta_t$ denote the base learning rate at iteration $t$, which is compatible with annealing techniques (e.g., cosine decay). For layer $\ell$, D-Muon updates the non-hidden layers using AdamW with learning rate $\eta_t$, and the hidden layers parameters $W_\ell \in \mathbb{R}^{d_{\text{out}}^\ell \times d_{\text{in}}^\ell}$ (i.e., QK, VO, MLP) with a rescaled learning rate $0.2\eta_t \sqrt{\max(d_{\text{in}}^\ell, d_{\text{out}}^\ell)}$. LANTON further rescales the hidden-layer learning rate to

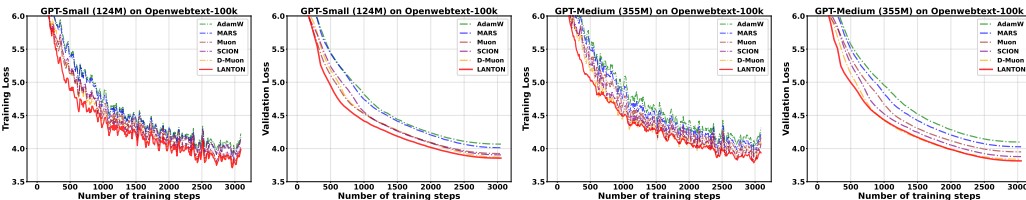

Figure 2: Training/validation loss on Openwebtext-100k datasets.

$0.2\eta_t\sqrt{\max(d^\ell_{\text{in}}, d^\ell_{\text{out}})\,\alpha^\ell_t/\alpha^m_t}$, where $\alpha^m_t = \max_{\ell \in \mathcal{G}_\ell} \alpha^\ell_t$ and $\mathcal{G}_\ell$ denotes the group of layer $\ell$. This is the practical instantiation of line 9 in Algorithm 1. In our implementation, there are three layer groups, i.e., {QK, VO, MLP}, {Embedding, LM-Head}, {LayerNorm}, so there are three noise factors $\alpha^m_t$ accordingly. For the first layer group (hidden layers), LANTON applies Newton-Schultz iterations with 5 steps (Jordan et al., 2024) to approximate the LMO update for matrix layers. For embedding and LM head layers, LANTON uses Signum (signed momentum) with a scaled base learning rate $r_1\,\eta_t$. For LayerNorm (vector) parameters, LANTON applies RMS-normalized updates with a scaled base learning rate $r_2\,\eta_t$. Similar to SCION, which requires two distinct update scales for layer groups, LANTON also specifies two update scales $r_1$ and $r_2$, with a base learning rate $\eta_t$.

### 6.2.2 GPT2 ON OPENWEBTEXT

We begin with small-scale experiments by pretraining GPT2 from scratch on OpenWebText-100k. All baselines (AdamW, MARS, Muon, SCION, D-Muon), and our method LANTON are trained for a single epoch with context length $512$ and batch size $16$. Unless otherwise specified, for all methods, we fix the random seed to $42$ and weight decay parameter $\gamma = 0.1$. We apply a cosine learning-rate schedule to the base step size $\eta_{\max}$ with a linear warmup of 300 steps. After warmup, the per-step learning rate is $\eta_t = \eta_{\min} + 1/2(\eta_{\max} - \eta_{\min})(1 + \cos(\frac{t\pi}{T}))$, where $t$ is the step index, $T$ is the number of training steps, and by default $\eta_{\min} = 0$. The detailed hyperparameter settings for every algorithm are summarized in 5 and Table 6 in Appendix H.

As shown in Figure 2, LANTON consistently dominates all baselines (AdamW, MARS, Muon, SCION, D-Muon). Its training loss drops fastest from the earliest iterations and stays below competing methods across the entire training, indicating superior convergence speed. LANTON also achieves the lowest validation loss, exhibit superior performance.

### 6.2.3 LLAMA ON C4 AND MINIPILE

We assess large-scale training by pretraining a LLaMA-1.1B model on C4 and a LLaMA-0.5B model on MiniPile with a total budget of 20B training tokens. We use the pretrained LLaMA tokenizer and set the sequence length to 256 on C4 and 512 on MiniPile. The batch size is 1024 for C4 and 300 for MiniPile. We employ a cosine learning rate schedule with a uniform warmup of 1,000 steps for all methods. Full hyperparameter settings for every baseline are reported in Tables 7 and 8 in Appendix H.

On C4, LANTON exhibits a significantly steeper loss descent in the early phase and maintains a consistent lead throughout training, while ultimately reaching validation losses comparable to other baselines (see Figure 3). We track the averaged effective learning rates within each layer group and provide the explanations for training acceleration of LANTON in Appendix K. On Minipile, although LANTON does not exhibit the lowest loss in the middle of training, it achieves the best final training loss and maintains consistently strong validation performance.

### 6.3 COMPARISON WITH ALGORITHMS USING LAYER-WISE/BLOCK-WISE LEARNING RATES

To highlight the benefit of our noise-adaptive layer-wise learning rate schedule, we compare with LAMB (You et al., 2019) and the recent block-wise scheme BW-AdamW (Wang et al., 2025). LAMB modifies Adam by applying a per-layer "trust ratio" to rescale the base learning rate in each layer. BW-AdamW manually tunes the best block-specific update ratio for each parameter block. Following the original best tuned ratio, we use $r(\text{Emb}) = 10, r(\text{QK}) = 8, r(\text{VO}) = 4, r(\text{MLP/LM-Head}) = 6, r(\text{Layer norm}) = 1$ in the experiment. The compared training and validation curves are presented in Figure 4(a). LANTON achieves much faster training speed with the

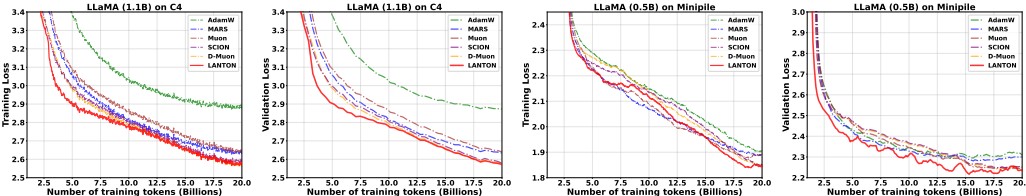

Figure 3: Training/validation loss on C4 and Minipile datasets.

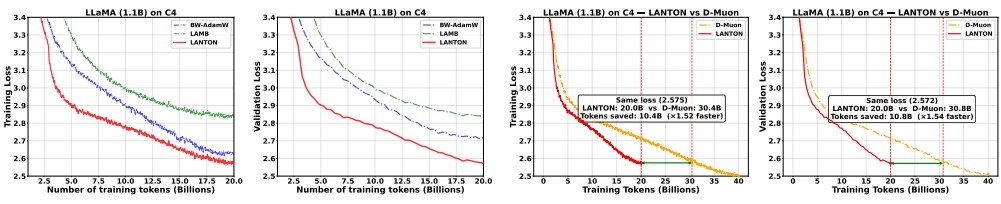

(a) Comparison with layer-/block-wise methods.  (b) Comparison of sample efficiency.

Figure 4: Training/validation loss on C4 datasets. (a) Comparison with algorithms using layer-wise/block-wise learning rates. (b) LANTON shows higher sample efficiency than D-Muon.

same budget of training tokens, and exhibits 0.1 lower validation loss than BW-AdamW. LANTON adapts the noise-adaptive layer-wise learning rate on the fly by monitoring gradient noise, whereas BW-AdamW uses fixed step sizes per parameter group. Moreover, neither baseline explicitly considers the parameter geometry properties.

## 6.4 SAMPLE EFFICIENCY WITH FIXED TOKEN BUDGET

To study the sample efficiency of our algorithm under various token budgets, we double the budget of tokens for D-Muon (i.e., 40B tokens) as that in LANTON (i.e., 20B tokens), and keep other experimental settings the same as that in Section 6.2.3, including the base learning rate, scale hyperparameters and batch size. Both algorithms use cosine learning rate decay, but the difference is that D-Muon has $2\times$ total training steps since it has $2\times$ more training tokens. Figure 4(b) shows that D-Muon and LANTON reach comparable training/validation losses when D-Muon uses about $1.5\times$ more tokens than LANTON (i.e., 30B tokens for D-Muon and 20B tokens for LANTON for reaching $\sim 2.57$ loss), demonstrating that the noise-adaptive learning rates can improve sample efficiency.

## 6.5 ROBUSTNESS TO BASE LEARNING RATE CHOICE

To evaluate sensitivity to the base learning rate, we keep the model (LLaMA-1.1B), dataset (C4), batch size (1024), optimizer settings, and cosine schedule fixed, then train LANTON with various base learning rates $\eta_{\max} \in \{0.001, 0.003, 0.005\}$. We compare against the best tuned D-MUON under the same setup. As shown in Figure 7 in Appendix I, we find that for all learning rates except for $\eta_{\max} = 0.001$, LANTON consistently achieves equal or lower loss with fewer training tokens, i.e., converges faster. With $\eta_{\max} = 0.001$, LANTON's loss still decreases faster for most (70%) of the training trajectory, with the two methods becoming close only toward the end. Overall, LANTON demonstrates robust performance across base learning rates and superior convergence speed in most hyperparameter settings.

## 7 CONCLUSION

We propose LANTON, a geometry-aware optimizer that incorporates noise-adaptive layer-wise learning-rate scaling on the top of LMO-based updates. By estimating gradient variance in the dual norm space and rescaling learning rate across layers, LANTON accelerates the transformer training hindered by heterogeneous and evolving noise. Theoretically, we obtain a sharp convergence rate of $\tilde{O}(1/\sqrt{T} + \sqrt{\sum_\ell \bar{\sigma}_\ell}/T^{1/4})$ with improved noise dependence across layers. Empirically, LANTON accelerates pretraining and improves validation metrics on GPT2 and LLaMA under a fixed token budget. One limitation of our work is that the theoretical results may depend on the parameter dimension. Another limitation is that our experiments are conducted on moderately sized models; extending and validating the approach at larger scales is an important direction for future work.

## REPRODUCIBILITY STATEMENT

We state the formal assumptions and results in the main text (Assumptions 5.1 and 5.2 and Theorem 5.3) and provide complete proofs of Theorem 5.3 in Appendices B and C. An anonymized code with training/evaluation scripts, configurations, seeds, and environment files is included in the supplementary materials. All base models are publicly available: LLaMA and GPT2-small/medium (used under their official research/community license; license text cited in Appendix N). Datasets C4, MiniPile, and OpenWebText are accessible on HuggingFace under the licenses stated on their corresponding Hugging Face dataset cards. We include download scripts, preprocessing/splits, and references to their dataset cards and licences (cited in Appendix N). These materials sufficiently support the reproduction of our results.

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

## A  TECHNICAL LEMMAS

In this section, we state several standard probabilistic and norm-equivalence lemmas without proof.

**Lemma A.1** (Azuma-Hoeffding inequality). *Let $\{Z_t\}_{t\geq 0}$ be a martingale with respect to filtration $\{\mathcal{F}_t\}_{t\geq 0}$. Assume that $|Z_t - Z_{t-1}| \leq c_t$ almost surely for all $t \geq 0$. Then for any fixed $T$, with probability at least $1 - \delta$,*

$$|Z_T - Z_0| \leq \sqrt{2 \sum_{t=1}^{T} c_t^2 \log(2/\delta)}.$$

**Lemma A.2** ((Liu et al., 2023c, Lemma 2.4)). *Suppose $X_1, \ldots, X_T$ is a martingale difference sequence adapted to a filtration $\mathcal{F}_1, \ldots, \mathcal{F}_T$ in a Hilbert space such that $\|X_t\|_F \leq R_t$ almost surely for some $R_t \geq 0$. Then for any $\delta \in (0, 1)$, with probability at least $1 - \delta$, for any fixed $t$ we have*

$$\left\| \sum_{s=1}^{t} X_s \right\|_F \leq 4 \sqrt{\log \frac{2}{\delta} \sum_{s=1}^{T} R_s^2}.$$

*Proof of Lemma A.2.* Since $\|\cdot\|_F$ satisfies $\|X + Y\|_F^2 \leq \|X\|_F^2 + \langle \nabla \|X\|_F^2, Y \rangle + \|Y\|_F^2$ for all $X, Y$, the condition for applying (Cutkosky & Mehta, 2021, Lemma 10) is satisfied, and therefore (Liu et al., 2023c, Lemma 2.4) holds. $\square$

**Lemma A.3** (Equivalence of norms). *For any two matrix norms $\|\cdot\|_a$ and $\|\cdot\|_b$, there exists $0 < C_1 \leq C_2$ (with $C_2 \geq 1$) such that $C_1 \|A\|_a \leq \|A\|_b \leq C_2 \|A\|_a$ for all matrices $A \in \mathbb{R}^{m \times n}$.*

*Remark A.4.* In the subsequent analysis, we will use the relationship among Frobenius norm $\|\cdot\|_F$, spectral norm $\|\cdot\|_2$, and nuclear norm $\|\cdot\|_{\text{nuc}}$. Specifically, for $A \in \mathbb{R}^{m \times n}$ we have

- $\|A\|_2 \leq \|A\|_F \leq \sqrt{\text{rank}(A)} \|A\|_2 \implies C_1 \leq 1, C_2 \geq \sqrt{\max\{m, n\}}$.

- $\|A\|_{\text{nuc}}/\sqrt{\text{rank}(A)} \leq \|A\|_F \leq \|A\|_{\text{nuc}} \implies C_1 \leq 1/\sqrt{\max\{m, n\}}, C_2 \geq 1$.

## B  PROOFS OF SECTION 5.1

We first recall a few key definitions from Equation (1) (with the convention $0/0 := 1$):

$$\kappa_\sigma^\ell = \begin{cases} \bar{\sigma}_\ell/\sigma_\ell & \sigma_\ell > 0 \\ 1 & \bar{\sigma}_\ell = 0 \end{cases}, \quad \kappa_\sigma = \max_\ell \kappa_\sigma^\ell, \quad \bar{\sigma}_{\max} = \max_\ell \bar{\sigma}_\ell, \quad \text{and} \quad t_0 = \frac{\log 2}{\log(1/\beta_2)}. \quad (5)$$

The following proofs are based on Assumptions 5.1 and 5.2 and the setting of Theorem 5.3. For simplicity, we omit the $\ell$ superscript/subscript whenever the context is clear.

**Lemma 5.4.** *With probability at least $1-\delta$, for all $\ell$ and $t_0 \leq t \leq T$, $\frac{\sigma_\ell^2(1-\beta_2^t)}{C_2} \leq H_t^\ell \leq 4\bar{\sigma}_\ell^2(1-\beta_2^t)$.*

*Proof of Lemma 5.4.* Consider the case where $0 < \underline{\sigma} \leq \bar{\sigma}$. Denote $c_{t,k} = \beta_2^{t-k}(1 - \beta_2)$. By Assumption 5.2 and Young's inequality,

$$H_t = \sum_{k=1}^{t} c_{t,k} \|G_k - \tilde{G}_k\|_*^2 \leq 2 \sum_{k=1}^{t} c_{t,k} \left( \|G_k - \nabla f(X_t)\|_*^2 + \|\tilde{G}_t - \nabla f(X_t)\|_*^2 \right)$$

$$\leq 4\bar{\sigma}^2 \sum_{k=1}^{t} c_{t,k} = 4\bar{\sigma}^2 \sum_{k=1}^{t} \beta_2^{t-k}(1 - \beta_2) = 4\bar{\sigma}^2(1 - \beta_2^t). \quad (6)$$

We proceed to derive high probability lower bound for $\sum_{k=1}^{t} c_{t,k} \|G_k - \tilde{G}_k\|_F^2$. Denote $\sigma_k^2 = \mathbb{E}_{k-1}[\|G_k - \nabla f(X_k)\|_F^2]$. Let $Z_k = c_{t,k}(\|G_k - \tilde{G}_k\|_F^2 - 2\sigma_k^2)$, then $\{Z_k\}_{k\geq 1}$ is a martingale difference sequence since

$$\mathbb{E}_{k-1}[Z_k] = \mathbb{E}_{t-1}[\|G_k - \tilde{G}_k\|_F^2 - 2\sigma_k^2]$$

$$= \mathbb{E}_{t-1}[\|G_k - \nabla f(X_k)\|_F^2 + \|\tilde{G}_k - \nabla f(X_k)\|_F^2 - 2\langle G_k - \nabla f(X_k), \tilde{G}_k - \nabla f(X_k)\rangle] - 2\sigma_k^2$$
$$= 0.$$

Using Assumption 5.2 and Lemma A.3 and Young's inequality, we have $Z_k \geq -2c_{t,k}\sigma_k^2$ and

$$Z_k \leq c_{t,k}\left(2C_2\left(\|G_k - \nabla f(X_k)\|_*^2 + \|\tilde{G}_k - \nabla f(X_k)\|_*^2\right) - 2\sigma_k^2\right) \leq c_{t,k}(4C_2\bar{\sigma}^2 - 2\sigma_k^2).$$

This implies that

$$|Z_k| \leq c_{t,k} \cdot \max\left\{2\sigma_k^2, 4C_2\bar{\sigma}^2 - 2\sigma_k^2\right\} = c_{t,k}(4C_2\bar{\sigma}^2 - 2\sigma_k^2),$$

where the last equality is due to $C_2 \geq 1$ and $\sigma_k \leq \bar{\sigma}$ almost surely. Then by the Azuma-Hoeffding inequality (Lemma A.1) and a union bound over $t$, for any $\delta \in (0,1)$, with probability at least $1 - \delta$, for all $t \leq T$,

$$\left|\sum_{k=1}^{t} Z_k\right| \leq \sqrt{2\sum_{k=1}^{t}(c_{t,k}(4C_2\bar{\sigma}^2 - 2\sigma_k^2))^2 \log\frac{2T}{\delta}} \leq (4C_2\bar{\sigma}^2 - 2\underline{\sigma}^2)\sqrt{\frac{2(1-\beta_2)}{1+\beta_2}\log\frac{2T}{\delta}}. \quad (7)$$

Rearranging Equation (7) yields that, with probability at least $1 - \delta$, for all $t \leq T$,

$$\sum_{k=1}^{t} c_{t,k}\|G_k - \tilde{G}_k\|_F^2 \geq 2\sum_{k=1}^{t} c_{t,k}\sigma_k^2 - (4C_2\bar{\sigma}^2 - 2\underline{\sigma}^2)\sqrt{\frac{2(1-\beta_2)}{1+\beta_2}\log\frac{2T}{\delta}}$$

$$\geq 2\underline{\sigma}^2(1-\beta_2^t) - (4C_2\bar{\sigma}^2 - 2\underline{\sigma}^2)\sqrt{\frac{2(1-\beta_2)}{1+\beta_2}\log\frac{2T}{\delta}}.$$

By the choice of $\beta_2$ in Theorem 5.3 and the definition of $t_0$, for all $t \geq t_0$ we have

$$\frac{4C_2\bar{\sigma}^2 - 2\underline{\sigma}^2}{\underline{\sigma}^2}\sqrt{\frac{2(1-\beta_2)}{1+\beta_2}\log\frac{2T}{\delta}} \leq \frac{1}{2} \quad \text{and} \quad (4C_2\bar{\sigma}^2 - 2\underline{\sigma}^2)\sqrt{\frac{2(1-\beta_2)}{1+\beta_2}\log\frac{2T}{\delta}} \leq \underline{\sigma}^2(1-\beta_2^t).$$

Therefore, by Lemma A.3, with probability at least $1 - \delta$, for all $t_0 \leq t \leq T$,

$$\sum_{k=1}^{t} c_{t,k}\|G_k - \tilde{G}_k\|_F^2 \geq \underline{\sigma}^2(1-\beta_2^t) \implies \sum_{k=1}^{t} c_{t,k}\|G_k - \tilde{G}_k\|_*^2 \geq \frac{\underline{\sigma}^2(1-\beta_2^t)}{C_2}. \quad (8)$$

We conclude the proof by combining Equations (6) and (8) and noting that the results also hold for the case $\underline{\sigma} = \bar{\sigma} = 0$. $\qquad\square$

**Lemma 5.5.** *With probability at least $1 - \delta$, for all $\ell$ and $t \leq T$,*

$$\min\left\{\frac{\alpha}{\sqrt{\alpha^2 + 4\bar{\sigma}_{\max}^2}}, \frac{1}{2\sqrt{C_2}\kappa_\sigma}\right\} =: \alpha_r \leq \frac{\alpha_t^\ell}{\alpha_t^m} \leq 1, \quad (2)$$

*and therefore, with probability at least $1 - \delta$, we have $\alpha_r\eta_{\min} \leq \eta_t^\ell \leq \eta_{\max}$ for all $\ell$ and $t \leq T$.*

*Proof of Lemma 5.5.* By Lemma 5.4, for all $t_0 \leq t \leq T$, it holds with probability at least $1 - \delta$ that

$$\frac{\underline{\sigma}^2(1-\beta_2^t)}{C_2} \leq \sum_{k=1}^{t} \beta_2^{t-k}(1-\beta_2)\|G_k - \tilde{G}_k\|_*^2 \leq 4\bar{\sigma}^2(1-\beta_2^t).$$

Therefore, with probability at least $1 - \delta$, for all $\ell$ and $t \leq T$,

$$\frac{\alpha}{\sqrt{\alpha^2 + 4\bar{\sigma}^2(1-\beta_2^t)}} \leq \alpha_t^\ell \leq \mathbb{I}(t < t_0) + \frac{\alpha}{\sqrt{\alpha^2 + \bar{\sigma}^2(1-\beta_2^t)/C_2}}\mathbb{I}(t \geq t_0). \quad (9)$$

Using Equation (9), we have

$$\frac{\alpha_t^\ell}{\alpha_t^m} \geq \frac{\alpha}{\sqrt{\alpha^2 + 4\bar{\sigma}^2(1-\beta_2^t)}}\left(\mathbb{I}(t < t_0) + \frac{\alpha}{\sqrt{\alpha^2 + \underline{\sigma}^2(1-\beta_2^t)/C_2}}\mathbb{I}(t \geq t_0)\right)^{-1}$$

$$= \frac{\alpha}{\sqrt{\alpha^2 + 4\bar{\sigma}^2(1-\beta_2^t)}}\mathbb{I}(t < t_0) + \frac{\sqrt{\alpha^2 + \sigma^2(1-\beta_2^t)/C_2}}{\sqrt{\alpha^2 + 4\bar{\sigma}^2(1-\beta_2^t)}}\mathbb{I}(t \geq t_0)$$

$$\geq \frac{\alpha}{\sqrt{\alpha^2 + 4\bar{\sigma}^2(1-\beta_2^t)}}\mathbb{I}(t < t_0) + \frac{\sigma}{2\sqrt{C_2}\bar{\sigma}}\mathbb{I}(t \geq t_0) \geq \min\left\{\frac{\alpha}{\sqrt{\alpha^2 + 4\bar{\sigma}^2}}, \frac{\sigma}{2\sqrt{C_2}\bar{\sigma}}\right\},$$

that is (we add back the subscript $\ell$ here),

$$\min\left\{\frac{\alpha}{\sqrt{\alpha^2 + 4\bar{\sigma}_\ell^2}}, \frac{\sigma_\ell}{2\sqrt{C_2}\bar{\sigma}_\ell}\right\} =: \alpha_r^\ell \leq \frac{\alpha_t^\ell}{\alpha_t^m} \leq 1.$$

Let $\alpha_r = \min_\ell \alpha_r^\ell$, and recall the definitions of $\bar{\sigma}_{\max}$ and $\kappa_\sigma$ in Equation (5), then for all $\ell$,

$$\min\left\{\frac{\alpha}{\sqrt{\alpha^2 + 4\bar{\sigma}_{\max}^2}}, \frac{1}{2\sqrt{C_2}\kappa_\sigma}\right\} =: \alpha_r \leq \frac{\alpha_t^\ell}{\alpha_t^m} \leq 1,$$

which gives Equation (2). The proof is completed. $\qquad\square$

## C  PROOF OF THEOREM 5.3

Before proving Theorem 5.3, we first provide a descent lemma for Algorithm 1.

**Lemma C.1.** *For the update in Algorithm 1, we have*

$$f(X_{t+1}) \leq f(X_t) + \sum_{\ell=1}^p \left(-\eta_t^\ell \|\nabla_\ell f(X_t)\|_{(\ell)*} + 2\eta_t^\ell \|B_t^\ell - \nabla_\ell f(X_t)\|_{(\ell)*} + \frac{L_\ell}{2}(\eta_t^\ell)^2\right).$$

*Moreover, we have*

$$\sum_{t=1}^T \sum_{\ell=1}^p \eta_t^\ell \|\nabla_\ell f(X_t)\|_{(\ell)*} \leq f(X_1) - f^* + \sum_{t=1}^T \sum_{\ell=1}^p \left(2\eta_t^\ell \|B_t^\ell - \nabla_\ell f(X_t)\|_{(\ell)*} + \frac{L_\ell}{2}(\eta_t^\ell)^2\right).$$

*Proof of Lemma C.1.* Applying (Riabinin et al., 2025, Lemma 1) with $X = X_t$ and $Y = X_{t+1}$,

$$f(X_{t+1}) \leq f(X_t) + \langle \nabla f(X_t), X_{t+1} - X_t \rangle + \sum_{\ell=1}^p \frac{L_\ell}{2}\|X_{t+1}^\ell - X_t^\ell\|_{(\ell)}^2$$

$$= f(X_t) + \sum_{\ell=1}^p \left(\langle \nabla_\ell f(X_t), X_{t+1}^\ell - X_t^\ell \rangle + \frac{L_\ell}{2}(\eta_t^\ell)^2\right).$$

For the second term, using the update of $X_{t+1}^\ell$ and the Cauchy-Schwarz inequality we have

$$\langle \nabla_\ell f(X_t), X_{t+1}^\ell - X_t^\ell \rangle = \langle B_t^\ell, X_{t+1}^\ell - X_t^\ell \rangle + \langle \nabla_\ell f(X_t) - B_t^\ell, X_{t+1}^\ell - X_t^\ell \rangle$$

$$\leq -\eta_t^\ell \|B_t^\ell\|_{(\ell)*} + \eta_t^\ell \|\nabla_\ell f(X_t) - B_t^\ell\|_{(\ell)*}$$

$$\leq -\eta_t^\ell \|\nabla_\ell f(X_t)\|_{(\ell)*} + 2\eta_t^\ell \|B_t^\ell - \nabla_\ell f(X_t)\|_{(\ell)*}.$$

Therefore, we obtain

$$f(X_{t+1}) \leq f(X_t) + \sum_{\ell=1}^p \left(-\eta_t^\ell \|\nabla_\ell f(X_t)\|_{(\ell)*} + 2\eta_t^\ell \|B_t^\ell - \nabla_\ell f(X_t)\|_{(\ell)*} + \frac{L_\ell}{2}(\eta_t^\ell)^2\right).$$

Rearranging the terms and taking summation over $t$ gives the result. $\qquad\square$

**Theorem 5.3.** *Suppose Assumptions 5.1 and 5.2 hold. Let $\Delta_1 = f(X_1) - f^*$. Set $\beta_1 = 1 - \alpha$ with $\alpha = \min\left(\frac{\sqrt{\Delta_1 \sum_\ell L_\ell}}{\sum_\ell \bar{\sigma}_\ell \sqrt{T}}, 1\right)$, $1 - \min_\ell \frac{\sigma_\ell^4}{32(2C_2\bar{\sigma}_\ell^2 - \sigma_\ell^2)^2 \log(4T/\delta)} \leq \beta_2 < 1$, $\eta_{\max} = \sqrt{\frac{\Delta_1 \alpha}{\sum_\ell L_\ell T}}$, and $\eta_{\min} = \eta_{\max}/\kappa_\eta$ with $1 \leq \kappa_\eta \leq O(1)$. With probability at least $1 - \delta$, we have*

$$\frac{1}{T}\sum_{t=1}^T \sum_{\ell=1}^p \|\nabla_\ell f(X_t)\|_{(\ell)*} \lesssim \frac{\sqrt{C_2}(\sum_\ell \bar{\sigma}_\ell)^2}{\sqrt{\Delta_1 \sum_\ell L_\ell T}} + \frac{C_2^{3/2}}{C_1}\sqrt{\log\frac{T}{\delta}}\left(\frac{\sqrt{\Delta_1 \sum_\ell L_\ell}}{\sqrt{T}} + \frac{\sqrt{\sum_\ell \bar{\sigma}_\ell}(\Delta_1 \sum_\ell L_\ell)^{1/4}}{T^{1/4}}\right).$$

*Proof of Theorem 5.3.* Define $\hat{\epsilon}_t^\ell = B_t^\ell - \nabla_\ell f(X_t)$, $\epsilon_t^\ell = G_t^\ell - \nabla_\ell f(X_t)$, and $S(X, Y) = \nabla f(X) - \nabla f(Y)$. Check that

$$\hat{\epsilon}_{t+1}^\ell = \beta_1 \hat{\epsilon}_t^\ell + (1 - \beta_1)\epsilon_t^\ell + S(X_t^\ell, X_{t+1}^\ell)$$

$$= \beta_1^t \hat{\epsilon}_1^\ell + (1 - \beta_1) \sum_{\tau=0}^{t-1} \beta_1^\tau \epsilon_{t-\tau}^\ell + \sum_{\tau=0}^{t-1} \beta_1^\tau S(X_{t-\tau}^\ell, X_{t+1-\tau}^\ell).$$

Using $L$-smoothness, $\|S(X_t^\ell) - S(X_{t+1}^\ell)\|_{(\ell)*} \le L_\ell \|X_{t+1}^\ell - X_t^\ell\|_{(\ell)} = L_\ell \eta_t^\ell \|O_t^\ell\|_{(\ell)} = L_\ell \eta_t^\ell$, and $\eta_t^\ell \le \eta_{\max}$ by Lemma 5.5,

$$\|\hat{\epsilon}_{t+1}^\ell\|_{(\ell)*} \le \beta_1^t \|\hat{\epsilon}_1^\ell\|_{(\ell)*} + (1 - \beta_1) \left\| \sum_{\tau=0}^{t-1} \beta_1^\tau \epsilon_{t-\tau}^\ell \right\|_{(\ell)*} + \eta_{\max} L_\ell \sum_{\tau=0}^{t-1} \beta_1^\tau.$$

Applying Lemma A.2 with $R_\tau = C_2 \beta_1^\tau \bar{\sigma}_\ell$ since $\|\beta_1^\tau \epsilon_{t-\tau}^\ell\|_F \le C_2 \|\beta_1^\tau \epsilon_{t-\tau}^\ell\|_{(\ell)*} \le C_2 \beta_1^\tau \bar{\sigma}_\ell$, a union bound over $t$, and Lemma A.3, with probability at least $1 - \delta$, for all $t \le T$,

$$\left\| \sum_{\tau=0}^{t-1} \beta_1^\tau \epsilon_{t-\tau}^\ell \right\|_{(\ell)*} \le \frac{1}{C_1} \left\| \sum_{\tau=0}^{t-1} \beta_1^\tau \epsilon_{t-\tau}^\ell \right\|_F \le \frac{4}{C_1} \sqrt{\log \frac{2T}{\delta} \sum_{\tau=0}^{t-1} (C_2 \beta_1^\tau \bar{\sigma}_\ell)^2} \le \frac{4 C_2 \bar{\sigma}_\ell}{C_1} \sqrt{\frac{\log(2T/\delta)}{1 - \beta_1}}.$$

Therefore, observing that $\hat{\epsilon}_1^\ell = \epsilon_1^\ell$ and plugging in the concentration bound yields

$$\|\hat{\epsilon}_{t+1}^\ell\|_{(\ell)*} \le \beta_1^t \bar{\sigma}_\ell + \frac{4 C_2}{C_1} (1 - \beta_1) \bar{\sigma}_\ell \sqrt{\frac{\log(2T/\delta)}{1 - \beta_1}} + \frac{\eta_{\max} L_\ell}{1 - \beta_1}.$$

Taking summation, with probability at least $1 - \delta$ we have

$$\sum_{t=1}^T \|\hat{\epsilon}_t^\ell\|_{(\ell)*} \le \frac{\bar{\sigma}_\ell}{1 - \beta_1} + \frac{4 C_2}{C_1} T \sqrt{1 - \beta_1} \bar{\sigma}_\ell \sqrt{\log \frac{2T}{\delta}} + \frac{T \eta_{\max} L_\ell}{1 - \beta_1}. \tag{10}$$

Recall Lemma C.1 and the definitions of $\Delta_1$ and $\hat{\epsilon}_t^\ell$,

$$\sum_{t=1}^T \sum_{\ell=1}^p \eta_t^\ell \|\nabla_\ell f(X_t)\|_{(\ell)*} \le \Delta_1 + \sum_{t=1}^T \sum_{\ell=1}^p \left( 2\eta_t^\ell \|\hat{\epsilon}_t^\ell\|_{(\ell)*} + \frac{L_\ell}{2} (\eta_t^\ell)^2 \right).$$

By Lemma 5.5 and a union bound (with Equation (10)), with probability at least $1 - 2\delta$,

$$\sum_{t=1}^T \sum_{\ell=1}^p \|\nabla_\ell f(X_t)\|_{(\ell)*} \le \frac{\Delta_1}{\alpha_r \eta_{\min}} + \sum_{\ell=1}^p \left( \frac{2\eta_{\max}}{\alpha_r \eta_{\min}} \sum_{t=1}^T \|\nabla_\ell f(X_t) - B_t^\ell\| + \frac{\eta_{\max}^2}{2\alpha_r \eta_{\min}} L_\ell T \right)$$

$$\le \frac{\kappa_\eta \Delta_1}{\alpha_r \eta_{\max}} + \sum_{\ell=1}^p \left( 2\kappa_\eta \left( \frac{\bar{\sigma}_\ell}{1 - \beta_1} + \frac{4 C_2}{C_1} T \sqrt{1 - \beta_1} \bar{\sigma}_\ell \sqrt{\log \frac{2T}{\delta}} \right) + \frac{\kappa_\eta \eta_{\max}}{\alpha_r} \left( \frac{2 T L_\ell}{1 - \beta_1} + \frac{L_\ell T}{2} \right) \right)$$

$$\le \frac{\kappa_\eta \Delta_1}{\alpha_r \eta_{\max}} + \frac{2\kappa_\eta}{\alpha_r} \left( \frac{\sum_\ell \bar{\sigma}_\ell}{1 - \beta_1} + \frac{4 C_2}{C_1} T \sqrt{1 - \beta_1} \sum_\ell \bar{\sigma}_\ell \sqrt{\log \frac{2T}{\delta}} \right) + \frac{5\kappa_\eta \eta_{\max} T \sum_\ell L_\ell}{\alpha_r (1 - \beta_1)}$$

$$\le \frac{6\kappa_\eta}{\alpha_r} \sqrt{\frac{\Delta_1 \sum_\ell L_\ell T}{1 - \beta_1}} + \frac{2\kappa_\eta}{\alpha_r} \left( \frac{\sum_\ell \bar{\sigma}_\ell}{1 - \beta_1} + \frac{4 C_2}{C_1} T \sqrt{1 - \beta_1} \sum_\ell \bar{\sigma}_\ell \sqrt{\log \frac{2T}{\delta}} \right)$$

$$\le \left( \frac{6\kappa_\eta}{\alpha_r} + \frac{2\kappa_\eta}{\alpha_r} \left( 1 + \frac{4 C_2}{C_1} \sqrt{\log \frac{2T}{\delta}} \right) \right) \sqrt{\Delta_1 \sum_\ell L_\ell T} + \frac{2\kappa_\eta (\sum_\ell \bar{\sigma}_\ell)^2 \sqrt{T}}{\alpha_r \sqrt{\Delta_1 \sum_\ell L_\ell}}$$

$$+ \left( \frac{6\kappa_\eta}{\alpha_r} + \frac{8 C_2 \kappa_\eta}{C_1 \alpha_r} \sqrt{\log \frac{2T}{\delta}} \right) \sqrt{\sum_\ell \bar{\sigma}_\ell} \left( \Delta_1 \sum_\ell L_\ell \right)^{1/4} T^{3/4},$$

where the last two inequalities use the choice of $\eta_{\max}$ and $\beta_1$ as stated in Theorem 5.3. Therefore, we obtain with probability at least $1 - 2\delta$ that

$$\frac{1}{T} \sum_{t=1}^T \sum_{\ell=1}^p \|\nabla_\ell f(X_t)\|_{(\ell)*} \le \left( \frac{6\kappa_\eta}{\alpha_r} + \frac{2\kappa_\eta}{\alpha_r} \left( 1 + \frac{4 C_2}{C_1} \sqrt{\log \frac{2T}{\delta}} \right) \right) \frac{\sqrt{\Delta_1 \sum_\ell L_\ell}}{\sqrt{T}} + \frac{2\kappa_\eta (\sum_\ell \bar{\sigma}_\ell)^2}{\alpha_r \sqrt{\Delta_1 \sum_\ell L_\ell T}}$$

$$+ \left( \frac{6\kappa_\eta}{\alpha_r} + \frac{8C_2\kappa_\eta}{C_1\alpha_r} \sqrt{\log \frac{2T}{\delta}} \right) \frac{\sqrt{\sum_\ell \bar{\sigma}_\ell}(\Delta_1 \sum_\ell L_\ell)^{1/4}}{T^{1/4}}.$$

Recall the definition of $\kappa_\sigma$ and $\alpha_r$ in Equations (2) and (5), with probability at least $1 - 2\delta$,

$$\frac{1}{T}\sum_{t=1}^{T}\sum_{\ell=1}^{p} \|\nabla_\ell f(X_t)\|_{(\ell)*} \leq \kappa_\eta \max \left\{ \sqrt{1 + \frac{4\bar{\sigma}_{\max}^2}{\alpha^2}}, 2\sqrt{C_2}\kappa_\sigma \right\} \left( \left( 8 + \frac{8C_2}{C_1}\sqrt{\log \frac{2T}{\delta}} \right) \frac{\sqrt{\Delta_1 \sum_\ell L_\ell}}{\sqrt{T}} \right.$$

$$\left. + \frac{2(\sum_\ell \bar{\sigma}_\ell)^2}{\sqrt{\Delta_1 \sum_\ell L_\ell T}} + \left( 6 + \frac{8C_2}{C_1}\sqrt{\log \frac{2T}{\delta}} \right) \frac{\sqrt{\sum_\ell \bar{\sigma}_\ell}(\Delta_1 \sum_\ell L_\ell)^{1/4}}{T^{1/4}} \right).$$

Replacing $\delta$ with $\delta/2$ completes the proof. $\qquad\square$

## D   EXPERIMENT OF IMAGE CLASSIFICATION

Following airbench setting in https://github.com/KellerJordan/cifar10-airbench and https://github.com/LIONS-EPFL/scion/tree/main/examples/airbench, we evaluate LANTON on CIFAR-100 image classification using an 8-layer convolutional neural network (CNN). Since stochastic gradient descent (SGD) generally outperforms AdamW on vision tasks, we follow the prior airbench setup and apply SGD to the norm and bias parameters for both Muon and D-Muon. LANTON partitions the parameters into two groups: (1) convolutional layers (matrix parameters), and (2) norm-layer and bias parameters. Newton–Schulz iterations are applied to the convolutional layers, while sign momentum is used for the norm and bias parameters. The full hyperparameter configuration is provided in Table XXX of the appendix.

As shown in Figure 5, all optimizers eventually reach nearly $100\%$ training accuracy on airbench CIFAR-100. However, LANTON exhibits a significantly faster convergence rate than other baselines: it reaches almost maximal training accuracy by around 70 epochs. More importantly, LANTON consistently achieves the highest validation accuracy, demonstrating that LANTON not only accelerates optimization throughout the training process but also yields superior generalization performance compared to all baselines.

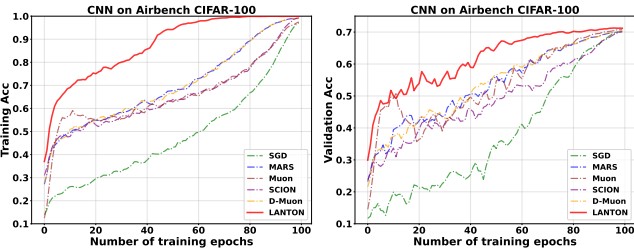

Figure 5: Training/validation accuracy on CIFAR-100.

Table 2: The hyperparameter settings in image classification.

| Method | $\eta_{\max}$ | Moment |
|--------|---------------|--------|
| SGD | 0.1 | $\beta = 0.85$ |
| Muon | 0.24 | $\beta_1 = 0.6, \beta_2 = 0.85, \beta_3 = 0.95$ |
| MARS | 0.1 | $\beta_1 = 0.9, \beta_2 = 0.95$ |
| SCION | 0.05 | $\beta = 0.5$ |
| D-Muon | 0.1 | $\beta_1 = 0.9, \beta_2 = 0.95, \beta_3 = 0.95$ |
| LANTON | 0.1 | $\beta_1 = 0.6, \beta_2 = 0.85$ |

## E   COMPARISON WITH ADAPTIVE VARIANT OF MUON

We additionally compared our method with the recently proposed adaptive variant AdaMuon (Si et al., 2025). Unlike LANTON, AdaMuon does not perform gradient noise estimation; instead, it introduces a momentum-style adaptive scaling on top of Muon and therefore is not noise-adaptive.

In our experiments, AdaMuon achieves slightly better performance than the original Muon but remains worse than LANTON. This matches our design motivation: LANTON is explicitly gradient noise-adaptive, adjusting each layer's learning rate based on its noise level. AdaMuon does not estimate noise and only plug a second-momentum term to Muon, providing limited gains.

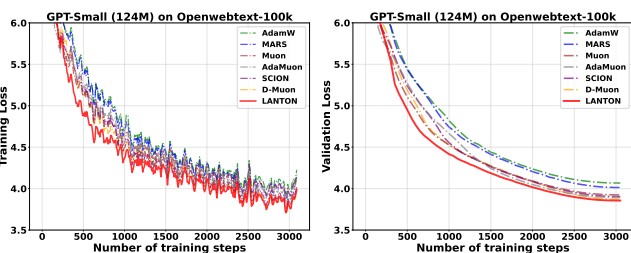

Figure 6: Training and validation loss on Openwebtext-100k.

## F    NOISE HETEROGENEITY

### F.1    IMPLEMENTATION DETAILS OF FIGURE 1

In this section, we provide implementation details of Figure 1. We pretrain LLaMA-1.1B model on C4 dataset for 10k steps, and apply momentum orthogonalized update to the matrix parameters $W_\ell \in \mathbb{R}^{d_{\text{out}} \times d_{\text{in}}}$ in the hidden layers (Query, Key, Value, MLP) and AdamW optimizer to the embedding and last layers. We first estimate gradient noise for two parameter groups, formed by matrix shape. For each weight matrix, we compute $\max(d_{\text{out}}, d_{\text{in}})$ and bucket it accordingly. We then aggregate the gradient-noise measure within each bucket over training (e.g., averaging across parameters in the group at each iteration) to obtain group-wise trajectories, which is shown in subfigure 1. Then we measure the layer-wise gradient noise within QK, VO, and MLP layer group in the last three subfigures.

The stochastic gradient noise is estimated by the nuclear norm (for parameters in Muon optimizer) or $\ell_1 \to \ell_1$ operator norm (for parameters in AdamW optimizer) of the difference between the current step's gradient and the previous step's gradient. The implementation follows Option I of line 7 in Algorithm 1 and line 4 in Table 1.

### F.2    NOISE MAGNITUDE ACROSS DIFFERENT LAYER GROUPS

We estimate the layer-wise gradient noise within the QK, VO, and MLP layer groups at the midpoint of training (5,000 steps). We find large layer-to-layer disparities within each group, indicating that gradient noise is far from uniform within a group. The statistics is presented in Table 3.

Table 3: The statistics of stochastic gradient noise in different layer groups of LLaMA.

| Layer Group | #Layers | $\bar{\sigma}$ | $\underline{\sigma}$ | $\sigma_{\text{mean}}$ |
|---|---|---|---|---|
| QK | 44 | 0.026 | 0.003 | 0.014 |
| VO | 44 | 0.117 | 0.009 | 0.046 |
| MLP | 66 | 0.107 | 0.018 | 0.038 |

## G    MODEL CONFIGURATIONS

We pretrain two types of model, GPT2 and LLaMA, the model configurations are listed in Table 4.

Table 4: Model configurations ($d_{\text{model}}$ denotes the hidden dimension, $d_{\text{FF}}$ denotes the feed-forward dimension, and $n_{\text{head}}$ denotes the number of attention head in transformer).

| Model | Size | $d_{\text{model}}$ | $d_{\text{FF}}$ | $n_{\text{head}}$ | depth |
|---|---|---|---|---|---|
| GPT-2 (small) | 124M | 768 | 3072 | 12 | 12 |
| GPT-2 (medium) | 355M | 1024 | 4096 | 16 | 24 |
| LLaMA (0.5B) | 522M | 1280 | 5120 | 20 | 15 |
| LLaMA (1.1B) | 1175M | 2048 | 5632 | 32 | 22 |

## H  HYPERPARAMETER SETTINGS

### H.1  HYPERPARAMETER SETTINGS IN GPT2 EXPERIMENTS

We tune the base learning rate $\eta_{\max}$ for each method via a grid search in the range of $[1 \times 10^{-4}, 5 \times 10^{-3}]$. For Muon baseline, we additionally sweep a separate base learning rate for non-hidden (embedding/output) layers. All runs use cosine decay from $\eta_{\max}$ down to $\eta_{\min} = 0.0$. Muon and D-Muon use three momentum hyperparameters: $(\beta_1, \beta_2)$ for the AdamW auxiliary optimizer and $\beta_3$ for orthogonalized momentum updates. LANTON uses two momentum parameters: $\beta_1$ for the gradient momentum and $\beta_2$ for the gradient noise momentum. All LMO-based methods (SCION, D-Muon, LANTON) apply layer-group learning-rate scaling; for SCION and D-Muon we adopt the best tuned scales reported in their original papers. All the hyperparameter settings are summarized in Table 5 and 6.

Table 5: The hyperparameter settings in GPT2-Small experiments.

| Method | $\eta_{\max}$ | Moment | Scale |
|---|---|---|---|
| AdamW | $1 \times 10^{-4}$ | $\beta_1 = 0.9, \beta_2 = 0.95$ | - |
| Muon | $(3 \times 10^{-3}, 3 \times 10^{-4})$ | $\beta_1 = 0.9, \beta_2 = 0.95, \beta_3 = 0.95$ | - |
| MARS | $1 \times 10^{-3}$ | $\beta_1 = 0.9, \beta_2 = 0.95$ | - |
| SCION | $3 \times 10^{-4}$ | $\beta = 0.9$ | $r_1 = 50, r_2 = 3000$ |
| D-Muon | $1 \times 10^{-3}$ | $\beta_1 = 0.9, \beta_2 = 0.95, \beta_3 = 0.95$ | $r = 0.2$ |
| LANTON | $5 \times 10^{-3}$ | $\beta_1 = 0.95, \beta_2 = 0.9$ | $r_1 = 300, r_2 = 1.0$ |

Table 6: The hyperparameter settings in GPT2-Medium experiments.

| Method | $\eta_{\max}$ | Moment | Scale |
|---|---|---|---|
| AdamW | $1 \times 10^{-4}$ | $\beta_1 = 0.9, \beta_2 = 0.95$ | - |
| Muon | $(3 \times 10^{-3}, 3 \times 10^{-4})$ | $\beta_1 = 0.9, \beta_2 = 0.95, \beta_3 = 0.95$ | - |
| MARS | $1 \times 10^{-3}$ | $\beta_1 = 0.9, \beta_2 = 0.95$ | - |
| SCION | $2 \times 10^{-4}$ | $\beta = 0.9$ | $r_1 = 50, r_2 = 3000$ |
| D-Muon | $5 \times 10^{-4}$ | $\beta_1 = 0.9, \beta_2 = 0.95, \beta_3 = 0.95$ | $r = 0.2$ |
| LANTON | $3 \times 10^{-3}$ | $\beta_1 = 0.95, \beta_2 = 0.9$ | $r_1 = 300, r_2 = 1.0$ |

### H.2  HYPERPARAMETER SETTINGS IN LLAMA EXPERIMENTS

The best base learning rate for each algorithm is grid searched over $\{1 \times 10^{-4}, 3 \times 10^{-4}, 5 \times 10^{-4}, 1 \times 10^{-3}, 3 \times 10^{-3}, 5 \times 10^{-3}\}$. The decayed layer rate is set as $\eta_{\min} = 1/10\eta_{\max}$ on C4 and $\eta_{\min} = 1/20\eta_{\max}$ on Minipile. We keep the momentum and scale parameters as that in GPT2 experiments. The hyperparameter choices on C4 and Minipile are summarized in Tables 7 and 8, respectively.

Table 7: The hyperparameter settings on C4.

| Method | $\eta_{\max}$ | $\eta_{\min}$ | Moment | Scale |
|---|---|---|---|---|
| AdamW | $3 \times 10^{-4}$ | $3 \times 10^{-5}$ | $\beta_1 = 0.9, \beta_2 = 0.95$ | - |
| Muon | $(5 \times 10^{-3}, 3 \times 10^{-4})$ | $(5 \times 10^{-4}, 3 \times 10^{-5})$ | $\beta_1 = 0.9, \beta_2 = 0.95, \beta_3 = 0.95$ | - |
| MARS | $1 \times 10^{-3}$ | $1 \times 10^{-4}$ | $\beta_1 = 0.9, \beta_2 = 0.95$ | - |
| SCION | $5 \times 10^{-4}$ | $5 \times 10^{-5}$ | $\beta = 0.9$ | $r_1 = 50, r_2 = 3000$ |
| D-Muon | $5 \times 10^{-3}$ | $5 \times 10^{-4}$ | $\beta_1 = 0.9, \beta_2 = 0.95, \beta_3 = 0.95$ | $r = 0.2$ |
| LANTON | $5 \times 10^{-3}$ | $5 \times 10^{-4}$ | $\beta_1 = 0.95, \beta_2 = 0.9$ | $r_1 = 300, r_2 = 1.0$ |

Table 8: The hyperparameter settings on Minipile.

| Method | $\eta_{\max}$ | $\eta_{\min}$ | Moment | Scale |
|---|---|---|---|---|
| AdamW | $8 \times 10^{-4}$ | $4 \times 10^{-5}$ | $\beta_1 = 0.9, \beta_2 = 0.95$ | - |
| Muon | $(5 \times 10^{-3}, 5 \times 10^{-4})$ | $(2.5 \times 10^{-4}, 2.5 \times 10^{-5})$ | $\beta_1 = 0.9, \beta_2 = 0.95, \beta_3 = 0.95$ | - |
| MARS | $1 \times 10^{-3}$ | $5 \times 10^{-5}$ | $\beta_1 = 0.9, \beta_2 = 0.95$ | - |
| SCION | $5 \times 10^{-4}$ | $2.5 \times 10^{-5}$ | $\beta = 0.9$ | $r_1 = 50, r_2 = 3000$ |
| D-Muon | $5 \times 10^{-3}$ | $2.5 \times 10^{-4}$ | $\beta_1 = 0.9, \beta_2 = 0.95, \beta_3 = 0.95$ | $r = 0.2$ |
| LANTON | $5 \times 10^{-3}$ | $2.5 \times 10^{-4}$ | $\beta_1 = 0.95, \beta_2 = 0.9$ | $r_1 = 300, r_2 = 1.0$ |

## I  ROBUSTNESS

The training and validation loss curves with different base learning rates are presented in Figure 7.

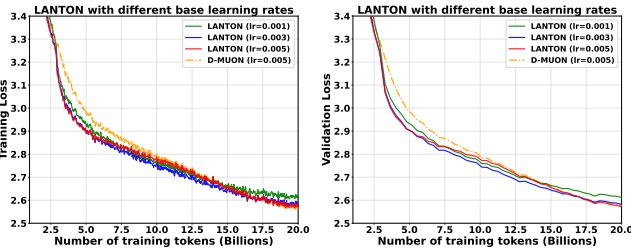

Figure 7: LANTON is robust to the choices of base learning rates.

## J  RUNNING TIME

To efficiently estimate the nuclear norm term $\|G_t^\ell - \tilde{G}_t^\ell\|_*^2$ for hidden-layer gradients (QK, VO, and MLP layers), we adopt the randomized singular value decomposition (R-SVD) method (Halko et al., 2011; Oh et al., 2015). The nuclear norm of a matrix is defined as the sum of its singular values, i.e., $\|A\|_* = \sum_i \sigma_i(A)$, where $\sigma_i(A)$ denotes the $i$-th singular value. Instead of computing a full SVD, we project $A = G_t^\ell - \tilde{G}_t^\ell$ onto a randomly generated low-dimensional subspace (with empirical dimension $d = 100$) and perform a small SVD on this reduced matrix to estimate its leading singular values. This approximation strategy is also used in SCION Pethick et al. (2025) in their implementation `https://github.com/LIONS-EPFL/scion/blob/main/examples/airbench/airbench_muon.py#L163`. The detailed implementation is provided in lines 44–70 of the submitted code file `train_llama/lanton.py`.

To further reduce computational cost, the gradient-noise estimation step (line 7 in Algorithm 1) is executed once every 10 iterations. We benchmark the runtime over 10 steps against other baselines, and the average computation time of each method is summarized in Table 9. Compared with the state-of-the-art baseline D-Muon, LANTON requires roughly 3 additional seconds per 10 iterations for gradient-noise estimation, resulting in an extra 0.84 hours of total training time. This corresponds to an overhead of only about $4\%$ relative to D-Muon. Moreover, Figure 8(a) reports the wall-clock runtime comparison. As shown, LANTON achieves a noticeably faster early loss decrease and then maintains a trajectory comparable to D-Muon for the remainder of training. These results demon-

strate that our method introduces only negligible computational overhead and achieves runtime on par with the SOTA baseline.

Table 9: The comparison of running time.

| Method | Time (second)/10 steps | Total running time (hours) |
|--------|------------------------|----------------------------|
| AdamW | 64.55 | 18.53 |
| Muon | 69.62 | 19.96 |
| MARS | 69.01 | 19.78 |
| SCION | 71.53 | 20.49 |
| D-Muon | 70.07 | 20.08 |
| LANTON | 73.08 | 20.92 |

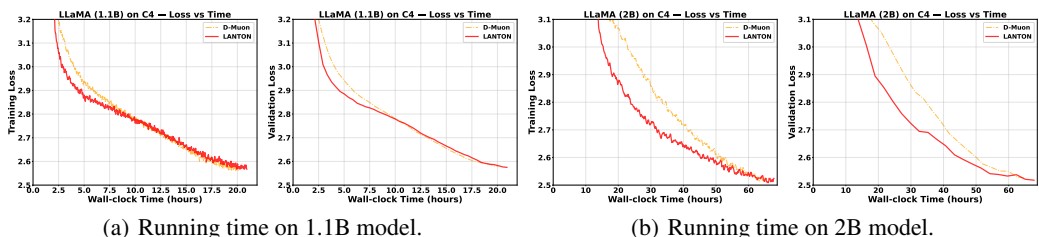

(a) Running time on 1.1B model.  (b) Running time on 2B model.

Figure 8: Training and validation loss vs. wall-clock time.

## K  EVOLUTION OF EFFECTIVE LEARNING RATE

The early-stage speedup arises because gradient noise varies significantly across layers at the beginning of training. As shown in Figure 9, the hidden layers (in subfigure (a)) start with an averaged effective learning-rate mean of $0.0028$ and a standard deviation of $0.0007$, indicating notable layerwise differences that LANTON can exploit to accelerate optimization in the early stage. By the end of training, cosine decay drives all learning rates toward very small values, and the hidden-layer learning rates converge to a mean of $0.00016$ with a much smaller standard deviation of $0.00008$. The reduced variance shows that layerwise learning rates become nearly uniform in the later stage of the training, and therefore layerwise learning rate is equivalent to using the same learning rate in the same group and the benefit diminishes.

Importantly, LANTON achieves faster early loss descent while still reaching comparable or better final performance, demonstrating that its advantage to accelerate training with noise-adaptive layerwise learning rates.

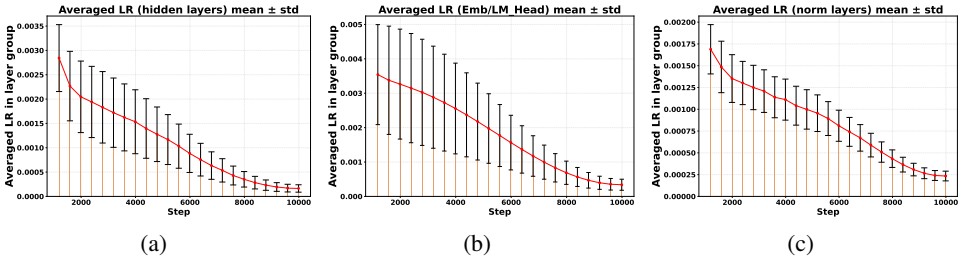

(a)  (b)  (c)

Figure 9: The statistics of learning rate in 3 layer groups: (a) start: $0.0028 \pm 0.0007$, end: $0.00016 \pm 0.00008$; (b) start: $0.0035 \pm 0.0015$, end: $0.00034 \pm 0.00016$; (c) start: $0.0017 \pm 0.0003$, end: $0.0002 \pm 0.00006$.

## L  ABLATION OF BATCH SIZE

To assess the influence of batch size on stochastic gradient variance estimation, we trained GPT (124M) models on openwebtext-100k with batch sizes BS $= \{8, 16, 32, 48, 64\}$ for one epoch (the number of training tokens is fixed to 46 million). For each batch size, we independently tuned the learning rate to its best-performing values ($1.0 \times 10^{-2}$ for BS=8, $5.0 \times 10^{-3}$ for other BS settings), ensuring a fair comparison across different settings. As shown in training loss curve in Figure 10, smaller batches yield noisier trajectories while larger batches produce smoother curves, yet all settings converge to nearly the same final training and validation loss (approximately 4.0).

These results demonstrate that our method is highly robust to batch-size variation: the convergence behavior and final performance are reasonably good and consistent across a wide range of batch sizes. Among the configurations, BS $= 16$ provides the best model performance, which is used in the main experimental settings.

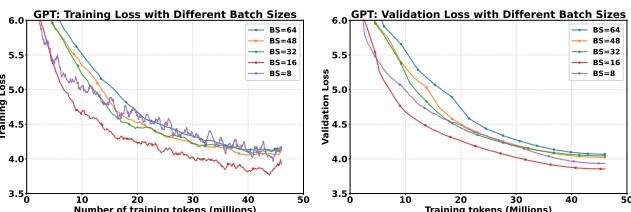

Figure 10: Training and validation loss vs. batch sizes (BS).

## M    GRADIENT NOISE ESTIMATION: OPTION I VS. OPTION II

We compared the performance of Options 1 and 2 in Algorithm 1. As described in line 7, our main experiments use Option 2 with GPT-124M on Openwebtext-100k. For Option 1, estimating gradient noise requires two independent mini-batches per iteration; therefore, under a fixed one-epoch budget, Option 1 performs only half as many optimization steps as Option 2.

Figure 11 reports the training and validation curves for both settings. With the same one-epoch budget, Option 1 achieves much lower final training and validation loss than Option 2 because it performs more gradient updates.

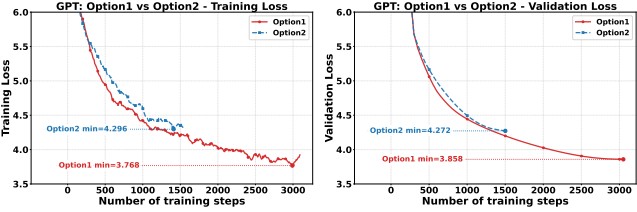

Figure 11: Training and validation loss with two gradient noise estimation options.

## N    LICENSE OF MODELS AND DATASETS

**GPT2**    OpenAI's GPT2 models are distributed by MIT License. We use only the open-source implementation of the GPT2 architecture in Hugging Face Transformers and do not redistribute Meta's model weights.

**LLaMA**    We follow Meta Llama 2 Community License Agreement. We use only the open-source implementation of the LLaMA architecture in Hugging Face Transformers and do not redistribute Meta's model weights.

**C4**    The English portion of the C4 (Colossal Clean Crawled Corpus) dataset comes from Hugging Face (allenai/c4), which is distributed under the Open Data Commons Attribution (ODC-By 1.0) license.

**Minipile**  It can be accessed from Hugging Face (JeanKaddour/minipile), which is distributed under MIT License.

**Openwebtext**  It can be accessed from Hugging Face (Skylion007/openwebtext), which is distributed under Creative Commons cc0-1.0 license.

## O  THE USE OF LARGE LANGUAGE MODELS (LLMS)

LLMs are not involved in our research methodology. Their use is limited to polish the writing.

