# OpenReview forum: "Noise-Adaptive Layerwise Learning Rates: Accelerating Geometry-Aware Optimization for Deep Neural Network Training"
_ICLR.cc/2026/Conference — Submitted to ICLR 2026_

### Official Review · Reviewer_w5Yy · 2025-10-17

**Soundness:** 2
**Presentation:** 2
**Contribution:** 2
**Rating:** 2
**Confidence:** 4

**Summary:**

The paper addresses inefficiencies in geometry-aware optimization algorithms for deep neural networks, which typically use fixed learning rates across all layers performing updates defined with respect to the same norm. Recognizing that local curvature and gradient noise can vary across layers and over training, the authors propose a noise-adaptive layerwise learning rate scheme. They call their method (being a combination of Scion [1]/Gluon [2] with this adaptive scheme) LANTON. This method dynamically estimates gradient variance associated with each layer's norm-constrained update and adjusts learning rates online. The authors provide convergence guarantees and present experiments on transformer models (LLaMA and GPT), which demonstrate faster training compared to some existing optimizers.

[1] Thomas Pethick, Wanyun Xie, Kimon Antonakopoulos, Zhenyu Zhu, Antonio Silveti-Falls, and Volkan Cevher. Training deep learning models with norm-constrained lmos. arXiv preprint arXiv:2502.07529, 2025.

[2] Artem Riabinin, Egor Shulgin, Kaja Gruntkowska, and Peter Richtarik. Gluon: Making muon \& Scion great again!(Bridging theory and practice of LMO-based optimizers for LLMs). arXiv preprint arXiv:2505.13416, 2025.

**Strengths:**

- The work addresses a research area that is active and highly relevant today.

- The first 3 sections of the paper are generally well-written.

- The method proposed in the paper represents a sound idea, as evidenced by good empirical performance relative to several baseline optimizers.

**Weaknesses:**

While the idea behind the proposed learning rate scheme is promising, I have several serious concerns about this submission. The first half of the paper is clearly written, but issues arise once the authors introduce their method. These include undefined or incorrectly used notation, lack of acknowledgment of prior work, and unrealistic assumptions. In particular:

1. The main update rule in LANTON (line 10, Algorithm 1) is essentially identical to the update rule of Scion [1]/Gluon [2]. Hence, the main contribution of the paper is the introduction of an adaptive layer-wise learning rate scheme rather than an entirely new algorithm. The paper does not clearly acknowledge that the main algorithmic components are inherited from prior work. Proper citations to [1], [2], and [3] should be included in the "Algorithmic Framework" paragraph of Section 4.

2. There is a **substantial overlap with [2], which the authors do not acknowledge**:

    - Lines 44–45 claim that "existing geometry-aware optimizers simply assign fixed learning rates within groups of layers associated with the same norm choice". This is inaccurate: [2] introduced Gluon, a family of layerwise LMO-based optimizers (including Muon and Scion as special cases) that already use different learning rates across layers (see Theorems 1 and 2 in [2]). Meanwhile, [2] is cited only once in the main part of this paper (for the smoothness assumption), completely ignoring other contributions.

    - The notation in Assumption 5.1 is taken directly from [2], but simplified: [2] uses generalized smoothness, while this paper only assumes classical (layer-wise) smoothness. In Section 5, the paper switches from a single matrix space to a product space without explaining the setup (e.g., the gradient component notation $\nabla_l f(X)$ is not defined). The paper does not properly introduce this notation. In line 139, the authors say the optimization is over $X \in \mathbb{R}^{m \times n}$, but it is actually over a product space of matrices, as defined in the Introduction section in [2].

    - This (unintroduced) notation is used correctly in Assumption 5.1, but not in several other places in the paper, e.g., the algorithm. For example, the authors define $G_t^l = \nabla F(X_t^l;\xi_t^l)$, but it should be $\nabla_l F(X_t;\xi_t)$, i.e., the $l$th component of the gradient evaluated on $(X_t;\xi_t)$, not the gradient evaluated on the $l$th component of $X_t$; this is necessary for the dimensions to match ($F$ takes all the model parameters as input). The gradient must be computed on the full parameter vector $X_t$, not just the $l$th component $X_t^l$ due to the mechanics of backpropagation - see [2]. For the same reason, the same sample $\xi_t$ is used for the whole gradient, so the $l$ index on $\xi_t$ is wrong.

    - This misunderstanding of the setup introduces further errors. For example, in Theorem 5.3, expressions like $f(X_1^l)$ are incorrect because $f$ takes the full model $X$, not a single layer $X^l$.

    - Many proofs very closely follow [2]: Lemma C.1 replicates the first part of Theorem 8 in [2] exactly (simplified by setting $L_i^1 = 0$), and Theorem 5.3 follows closely the remainder of that proof, differing in the assumption on stochastic gradient noise. The mistake with $f(X_1^l)$ also appears here.

Overall, the authors not only borrow the setup and notation from [2] without acknowledgment, but also use it incorrectly. They claim their main contribution is being the first to introduce layer-specific learning rates in LMO-based optimizers, but this was already done in [2], which also proved results under more general assumptions (classical smoothness is known to be a poor model for neural network loss landscapes, while generalized smoothness better captures their behavior [4]).

3. I am very skeptical about the practicality of Assumption 5.2, which requires that
$$\underline{\sigma}_l\leq\|\nabla_l F(X,\xi) - \nabla_l f(X)\|_{(l)*} \leq\bar{\sigma}_l$$
with probability one across all layers. The uniform upper bound is already very restrictive, and the lower bound is generally unrealistic (for example, it is forced to be $0$ under interpolation). But when $\underline{\sigma}_l=0$, Theorem 5.3 requires that $1\leq \beta_2<1$, which is impossible.

[1] Thomas Pethick, Wanyun Xie, Kimon Antonakopoulos, Zhenyu Zhu, Antonio Silveti-Falls, and Volkan Cevher. Training deep learning models with norm-constrained lmos. arXiv preprint arXiv:2502.07529, 2025.

[2] Artem Riabinin, Egor Shulgin, Kaja Gruntkowska, and Peter Richtarik. Gluon: Making muon \& Scion great again!(Bridging theory and practice of LMO-based optimizers for LLMs). arXiv preprint arXiv:2505.13416, 2025.

[3] Keller Jordan, Yuchen Jin, Vlado Boza, You Jiacheng, Franz Cesista, Laker Newhouse, and Jeremy Bernstein. Muon: An optimizer for hidden layers in neural networks, 2024.

[4] Zhang, Jingzhao, Tianxing He, Suvrit Sra, and Ali Jadbabaie. Why gradient clipping accelerates training: A theoretical justification for adaptivity. arXiv preprint arXiv:1905.11881, 2019.

**Questions:**

I think that the overall idea behind the submission is good, but the paper itself has serious flaws that make it not publishable in the current state. The setup is not introduced, it completely ignores that large chunks of the theory are borrowed from another work, and it misuses some chunks of this work by making errors. Hence, I do not think it is po
Could the authors please address my comments above?

- How can Assumption 5.2 be justified beyond analytic convenience?

- How does the proposed algorithm compare with other adaptive variants of Muon?

**Details Of Ethics Concerns:**

The paper presents a promising idea; however, in its current form, it is not publishable. The authors fail to properly introduce key aspects of the setup, rely heavily on prior work without any acknowledgment, and make errors in the process. I also have two additional questions:

1. How can Assumption 5.2 be justified beyond analytic convenience?

2. In recent months, several adaptive variants of Muon have been proposed. Have the authors tested how their method compares empirically with these variants?

---

> ### Author Response · Authors · 2025-11-23
>
> **Q1. The main update rule in LANTON (line 10, Algorithm 1) is essentially identical to the update rule of Scion [1] and Gluon [2]. Hence, the main contribution of the paper is the introduction of an adaptive layer-wise learning rate scheme rather than an entirely new algorithm. The paper does not clearly acknowledge that the main algorithmic components are inherited from prior work. Proper citations to [1, 2, 3] should be included in the "Algorithmic Framework" paragraph of Section 4.**
>
> **A1.** Thank you for your suggestions. We have included the citations [1, 2, 3] in the Algorithmic Framework paragraph of Section 4 in the revised paper.
>
> **Q2. There is a substantial overlap with [2], which the authors do not acknowledge.**
>
> **A2.**
> - Thank you for pointing this out. We have added additional discussion and explicitly acknowledge the contributions of [2] -- particularly Gluon as a family of layerwise LMO-based optimizers that use different learning rates across layers -- in both the introduction and related work sections. We also clarify that [2] firstly proposed layerwise learning rates for the geometry-aware optimization methods based on smoothness parameters. In contrast, we focus on the heterogeneous noise magnitude of each layer instead of the smoothness parameters, and estimate the gradient variance on the fly then use it to assign time-varying, noise-adaptive layerwise learning rates within each group.
>
> - We acknowledge that Assumption 5.1 simplifies the generalized layer-wise smoothness introduced in [2] to classical layer-wise smoothness, but this is only to make the presentation cleaner. The main focus of this work is the noise-adaptive learning rate scheme.
>
> - We have revised Sections 3, 4 and Theorem 5.3 to ensure that our notation is fully consistent throughout the paper.
>
> - We would like to clarify that the main technical challenge in the proof is handling the layerwise noise-adaptive component (lines 7–8 in Algorithm 1) and establishing high-probability guarantees for the variance tracker $H_t^{\ell}$ and the noise ratio term $\alpha_t^{\ell}/\alpha_t^{m}$ (see Lemmas 5.4 and 5.5). In addition, we would like to emphasize that we have not overstated the novelty of the remaining parts of our proof. We have already cited [2] in Lemma C.1. Moreover, our proof sketch for Theorem 5.3 primarily references [4] because recent convergence analyses of Muon [2, 5, 6] largely follow the momentum-normalization framework introduced in [4, Theorem 1]. We have added explicit citations to these works [2, 5, 6] in Section 5 of the revised paper to ensure full acknowledgment of relevant recent literature.
>
> - The focus of our paper is to design **noise-adaptive learning rate** instead of the generalized smoothness model. They are orthogonal to each other. We have already mentioned it in the revised version of introduction.
>
> [1] Thomas Pethick, Wanyun Xie, Kimon Antonakopoulos, Zhenyu Zhu, Antonio Silveti-Falls, and Volkan Cevher. Training deep learning models with norm-constrained lmos. arXiv preprint arXiv:2502.07529, 2025.
>
> [2] Artem Riabinin, Egor Shulgin, Kaja Gruntkowska, and Peter Richt´arik. Gluon: Making muon & scion great again!(bridging theory and practice of lmo-based optimizers for llms). arXiv preprint arXiv:2505.13416, 2025.
>
> [3] Keller Jordan, Yuchen Jin, Vlado Boza, You Jiacheng, Franz Cesista, Laker Newhouse, and Jeremy Bernstein. Muon: An optimizer for hidden layers in neural networks, 2024.
>
> [4] Ashok Cutkosky and Harsh Mehta. Momentum improves normalized sgd. In International Conference on Machine Learning, pp. 2260–2268. PMLR, 2020.
>
> [5] Jiaxiang Li and Mingyi Hong. A note on the convergence of muon. arXiv preprint arXiv:2502.02900, 2025.
>
> [6] Wei Shen, Ruichuan Huang, Minhui Huang, Cong Shen, and Jiawei Zhang. On the convergence analysis of muon. arXiv preprint arXiv:2505.23737, 2025

---

> ### Author Response · Authors · 2025-11-23
>
> **Q3. I am very skeptical about the practicality of Assumption 5.2, which requires that $\underline{\sigma}\_{\ell} \leq \\|\nabla_{\ell} F(X,\xi)-\nabla_{\ell} f(X)\\|\_{(\ell)*}\leq \bar{\sigma}\_{\ell}$ with probability one across all layers. The uniform upper bound is already very restrictive, and the lower bound is generally unrealistic (for example, it is forced to be 0 under interpolation). But when $\underline{\sigma}\_{\ell}=0$, Theorem 5.3 requires that $1\leq \beta_2<1$, which is impossible.**
>
> **How can Assumption 5.2 be justified beyond analytic convenience?**
>
> **A3.** First, we would like to emphasize that the almost-sure uniform upper and lower bounds in Assumption 5.2(ii) are assumed purely for technical reasons to enable high probability analysis and convergence guarantees.
>
> Second, Assumption 5.2(ii) can be empirically supported by Figure 1. As shown in the first subfigures of Figure 1, the gradient noise does not vanish across all interations during the training process. In practice, we can set $\underline{\sigma}\_{\ell}$ to the minimum gradient noise among all layers and $\bar{\sigma}\_{\ell}$ to the maximum, ensuring that $0< \underline{\sigma}\_{\ell} \leq \\|\nabla_{\ell} F(X,\xi)-\nabla_{\ell} f(X)\\|\_{(\ell)*}\leq \bar{\sigma}\_{\ell}$, consistent with the observations in Figure 1 (in fact, we rarely observe cases where $\bar{\sigma}\_{\ell}>0$ while $\underline{\sigma}\_{\ell}=0$ in practice). Under the noiseless setting, we simply have $\underline{\sigma}\_{\ell}= \bar{\sigma}\_{\ell}= 0$. We acknowledge that our current theory cannot handle the case where $\bar{\sigma}\_{\ell}>0$ and $\underline{\sigma}\_{\ell}=0$, but this is unlikely to happen in practice as mentioned above.
>
> Third, with the convention $0/0:=1$ (see Section 5.1), when $\underline{\sigma}\_{\ell}=0$, it necessarily implies $\underline{\sigma}\_{\ell}= \bar{\sigma}\_{\ell}= 0$ corresponding to the noiseless case. In this case, the requirement on $\beta_2$ in Theorem 5.3 simplifies to $0\leq \beta_2< 1$, which is always satisfied.
>
> **Q5. How does the proposed algorithm compare with other adaptive variants of Muon? (In recent months, several adaptive variants of Muon have been proposed. Have the authors tested how their method compares empirically with these variants?)**
>
> **A5.**
> We additionally compared our method with the recently proposed adaptive variant AdaMuon [1]. Unlike LANTON, AdaMuon does not perform gradient noise estimation; instead, it introduces a momentum-style adaptive scaling on top of Muon and therefore is not noise-adaptive.
>
> In our experiments shown in **Figure 6 in Appendix E**, AdaMuon achieves slightly better performance than the original Muon but remains worse than LANTON. This matches our design motivation: LANTON is explicitly gradient noise-adaptive, adjusting each layer's learning rate based on its noise level. AdaMuon does not estimate noise and only plug a second-momentum term to Muon, providing limited gains.
>
> [1] Chongjie Si, Debing Zhang, and Wei Shen. Adamuon: Adaptive muon optimizer. arXiv e-prints, pp. arXiv–2507, 2025.

---

### Official Review · Reviewer_wJCs · 2025-10-29

**Soundness:** 3
**Presentation:** 2
**Contribution:** 3
**Rating:** 4
**Confidence:** 5

**Summary:**

This paper finds that existing geometry-aware optimizers apply a single, fixed learning rate to groups of layers that share the same norm. However, the stochastic gradient noise could be highly heterogeneous _within_ these groups and dynamic throughout training, rendering a uniform learning rate suboptimal.
To address this, the paper proposes LANTON, a layer-wise adaptive learning rate schedule built on top of existing LMO-based optimizers (Scion). Its mechanism estimates the gradient variance for each layer in the dual norm space to scale that layer's learning rate. The principle is to assign smaller step sizes to layers with higher estimated gradient noise.
The authors provide a theoretical convergence analysis, claiming an improved rate compared with a uniform maximum method. Empirically, they demonstrate that LANTON accelerates the training of GPT2 and LLaMA models, achieving faster convergence and showing.

**Strengths:**

1. The motivation is clear. This paper is motivated by (Wang et al., 2025), which notices different loss curvature on different layers of Adam. This paper empirically shows a similar observation on Muon.
2. This paper applies the noisy adaptive idea within the geometry-aware LMO framework by estimating the variance in the dual norm space.
3. LANTON shows an improved convergence rate compared with Scion using a uniform learning rate.
4. The paper shows consistent improvement on both GPT2 and LLaMA models.

**Weaknesses:**

1. This paper doesn't discuss about computational overhead. LANTON needs to compute the dual norm additionally. Especially for hidden layers, it requires extra nuclear norm computation. I suppose the computation of the nuclear norm is not negligible. How to compute it in implementation? How much computational overhead does it require? Could you compare with other methods according to wall-clock time?
2. Miss image task comparison. This paper's experiments only focus on language tasks. I'm curious if LANTON can show improvement in image tasks like CIFAR-10?
3. I feel LANTON shows speedup at the beginning of training. Would the benefit diminish for long-term training?
4. I think Figure 4 (b) is an unfair comparison. If I understand correctly, Figure 4 (b) has the same setup as Figure 3 (left) except number of tokens / training steps. In Figure 3, LANTON shows a similar final performance to D-Muon. How could you claim a speedup if you run D-Muon 2$\times$ longer? Such speedup may be just brought by a different learning rate and its schedule.

**Questions:**

1. Do you compare the performance between options 1 and 2 in Algorithm 1?
2. I suppose y-axis of subfigures 2 and 4 in Figure 2 should be "Validation loss".
3. Is there any potential reason why D-Muon converges faster than LANTON in subfigure 4 of Figure 2?

---

> ### Author Response · Authors · 2025-11-23
>
> **Q1. This paper doesn't discuss about computational overhead. LANTON needs to compute the dual norm additionally. Especially for hidden layers, it requires extra nuclear norm computation. I suppose the computation of the nuclear norm is not negligible. How to compute it in implementation? How much computational overhead does it require? Could you compare with other methods according to wall-clock time?**
>
> **A1.** To efficiently estimate the nuclear norm term $\\|G\_{t}^{\ell}-\tilde{G}\_{t}^{\ell}\\|\_{*}^{2}$ for hidden-layer gradients (QK, VO, and MLP layers), we adopt the randomized singular value decomposition (R-SVD) method [1, 2].
>
> The nuclear norm of a matrix is defined as the sum of its singular values, i.e., $\\|A\\|\_* = \sum\_{i} \sigma\_i(A)$,
> where $\sigma\_i(A)$ denotes the $i$-th singular value.
> Instead of computing a full SVD, we project $A = G\_{t}^{\ell} - \tilde{G}\_{t}^{\ell}$ onto a randomly generated low-dimensional subspace (with empirical dimension $d = 100$) and perform a small SVD on this reduced matrix to estimate its leading singular values. This approximation strategy is also used in SCION \cite{pethick2025training} in their implementation https://github.com/LIONS-EPFL/scion/blob/main/examples/airbench/airbench_muon.py#L163. The detailed implementation is provided in lines 44-70 of the submitted code file *train\_llama/lanton.py*.
>
>
> To further reduce computational cost, the gradient-noise estimation step (line 7 in Algorithm 1) is executed once every 10 iterations. We benchmark the running time over 10 steps against other baselines, and the average computation time of each method is summarized in **Table 1**. Compared with the state-of-the-art (SOTA) baseline D-Muon, LANTON requires roughly 3 additional seconds per 10 iterations for gradient-noise estimation, resulting in an extra 0.84 hours of total training time. This corresponds to an overhead of only about $4\\%$ relative to D-Muon. Moreover, **Figure 8 in Appendix J** reports the wall-clock runtime comparison. As shown, LANTON achieves a noticeably faster early loss decrease and then maintains a trajectory comparable or better to D-Muon for the remainder of training. These results demonstrate that our method introduces only negligible computational overhead and achieves running time on par with the SOTA baseline.
>
> **Table 1: The comparison of running time on training 1.1B model.**
>
> | **Method** | **Time (second) / 10 steps** | **Total running time (hours)** |
> |-----------|-------------------------------|---------------------------------|
> | AdamW     | 64.55                         | 18.53                           |
> | Muon      | 69.62                         | 19.96                           |
> | MARS      | 69.01                         | 19.78                           |
> | SCION     | 71.53                         | 20.49                           |
> | D-Muon    | 70.07                         | 20.08                           |
> | LANTON    | 73.08                         | 20.92                           |
>
>
> [1] Nathan Halko, Per-Gunnar Martinsson, and Joel A Tropp. Finding structure with randomness: Probabilistic algorithms for constructing approximate matrix decompositions. SIAM review, 53 (2):217–288, 2011.
>
> [2] Tae-Hyun Oh, Yasuyuki Matsushita, Yu-Wing Tai, and In So Kweon. Fast randomized singular value thresholding for nuclear norm minimization. In Proceedings of the IEEE Conference on Computer Vision and Pattern Recognition, pp. 4484–4493, 2015.
>
> **Q2. Miss image task comparison. This paper's experiments only focus on language tasks. I'm curious if LANTON can show improvement in image tasks like CIFAR-10?**
>
> **A2.** Following airbench setting in https://github.com/KellerJordan/cifar10-airbench and https://github.com/LIONS-EPFL/scion/tree/main/examples/airbench, we evaluate LANTON on CIFAR-100 image classification using an 8-layer convolutional neural network (CNN). Since stochastic gradient descent (SGD) generally outperforms AdamW on vision tasks, we follow the prior airbench setup and apply SGD to the norm and bias parameters for both Muon and D-Muon.
> LANTON partitions the parameters into two groups: (1) convolutional layers (matrix parameters), and (2) norm-layer and bias parameters. Newton-Schulz iterations are applied to the convolutional layers, while sign momentum is used for the norm and bias parameters. The full hyperparameter configuration is provided in Table 2 of the appendix.
>
> As shown in **Figure 5 in Appendix D**, all optimizers eventually reach nearly $100\\%$ training accuracy on airbench CIFAR-100. However, LANTON reaches $99\\%$ training accuracy by around 70 epochs, while other baselines require at least 90 epochs. More importantly, LANTON consistently achieves the highest validation accuracy, demonstrating that LANTON not only accelerates optimization throughout the training process but also yields superior generalization performance compared to all baselines.

---

> ### Author Response · Authors · 2025-11-23
>
> **Q3. I feel LANTON shows speedup at the beginning of training. Would the benefit diminish for long-term training?**
>
> **A3.** The early-stage speedup arises because gradient noise varies significantly across layers at the beginning of training. As shown in **Figure 9 in Appendix K**, the hidden layers (in subfigure (a)) start with an averaged effective learning-rate mean of $0.0028$ and a standard deviation of $0.0007$, indicating notable layer-wise differences that LANTON can exploit to accelerate optimization in the early stage.
> By the end of training, cosine decay drives all learning rates toward very small values, and the hidden-layer learning rates converge to a mean of $0.00016$ with a much smaller standard deviation of $0.00008$. The reduced variance shows that layerwise learning rates become nearly uniform in the later stage of the training, and therefore layerwise learning rate is equivalent to using the same learning rate in the same group and the benefit diminishes.
>
> Importantly, LANTON achieves faster early loss descent while still reaching comparable or better final performance, demonstrating that its advantage to accelerate training with noise-adaptive layer-wise learning rates.
>
> **Q4. I think Figure 4 (b) is an unfair comparison. If I understand correctly, Figure 4 (b) has the same setup as Figure 3 (left) except number of tokens / training steps. In Figure 3, LANTON shows a similar final performance to D-Muon. How could you claim a speedup if you run D-Muon 2x longer? Such speedup may be just brought by a different learning rate and its schedule.**
>
> **A4.** Our intention in Figure 4 (b) was to highlight that LANTON reduces the sample complexity of training, i.e., it reaches
> the same loss level using fewer training tokens. We did not intend to claim a training speedup over D-Muon under different learning-rate schedules. We have revised the text in the revised submission to emphasize sample efficiency instead of the claim of speedup.
>
> **Q5. Do you compare the performance between options 1 and 2 in Algorithm 1?**
>
> **A5.** We compared the performance of Options 1 and 2 in Algorithm 1. As described in line 7, our main experiments use Option 2 with GPT-124M on Openwebtext-100k. For Option 1, estimating gradient noise requires two independent mini-batches per iteration; therefore, under a fixed one-epoch budget, Option 1 performs only half as many optimization steps as Option 2.
>
> **Figure 11 in Appendix M** reports the training and validation curves for both settings. With the same one-epoch budget, Option 1 achieves much lower final training and validation loss than Option 2 because it performs more gradient updates.
>
> **Q6. I suppose y-axis of subfigures 2 and 4 in Figure 2 should be "Validation loss".**
>
> **A6.** Your understanding is correct. Thank you for the helpful suggestion. We have updated the figures in the revised version.
>
> **Q7. Is there any potential reason why D-Muon converges faster than LANTON in subfigure 4 of Figure 2?**
>
> **A7.** We reused the optimal initial base learning rates in GPT-small to the GPT-medium, which may not be the best for the GPT-medium. So we tuned the best optimal hyperparameters for all the baselines in GPT-medium settings. We have updated the Figure 2 in the revised version, the corresponding tuned hyperparameters are shown in **Table 6 in Appendix H**.

---

### Official Review · Reviewer_7VGb · 2025-10-31

**Soundness:** 3
**Presentation:** 4
**Contribution:** 3
**Rating:** 8
**Confidence:** 3

**Summary:**

The paper proposes a layer-wise learning rate scaling mechanism that works with existing geometry-aware optimizers. The main idea is to dynamically choose smaller learning rates for layers with higher stochastic gradient variance, and larger ones for layers with lower variance, based on online variance estimation. Geometry awareness is maintained through the use of the Linear Minimization Oracle (LMO) for norm-constrained updates. Even among layers that share the same type (and thus the same norm/LMO), the gradient noise and curvature can differ across layers and change during training. However, existing geometry-aware optimizers typically assign a fixed learning rate to all layers within a group, ignoring this variation. The paper shows both theoretical and empirical improvements: the proposed method achieves a better convergence rate than fixed-rate geometry-aware methods and demonstrates faster convergence and up to 1.5× higher sample efficiency on GPT-2 and LLaMA-2 experiments.

**Strengths:**

The paper is clearly organized and easy to follow. I am not deeply familiar with geometry-aware optimization but enjoy reading the manuscript.

The method is well-motivated. LANTON effectively tackles the overlooked issue of intra-group heterogeneity in gradient noise.

The proposed approach is plug-and-play and integrates naturally with existing geometry-aware frameworks such as Muon, as well as common learning rate schedules.

The experiments on GPT-2 and LLaMA convincingly support the paper’s claims, demonstrating faster convergence and better sample efficiency.

The convergence analysis is interesting and well connected to the method. However I do not have the technical background to verify all the derivations.

**Weaknesses:**

I do not see any major weaknesses. The additional computational and memory overhead introduced by maintaining per-layer noise estimates is expected and acceptable.

Empirical estimation of stochastic gradient variance may depend strongly on batch size, but the paper does not include an analysis or sensitivity study in this regard. Including such an analysis would strengthen the empirical validation.

**Questions:**

The theoretical bound includes constants depending on parameter dimensions. Can the authors clarify how large these constants are in practice, and whether they affect scalability to larger models?

---

> ### Author Response · Authors · 2025-11-23
>
> We thank the reviewer for the insightful comments, and we have addressed your concerns as follows.
>
> **Q1. Empirical estimation of stochastic gradient variance may depend strongly on batch size, but the paper does not include an analysis or sensitivity study in this regard. Including such an analysis would strengthen the empirical validation.**
>
> **A1.** To assess the influence of batch size on stochastic gradient variance estimation, we trained GPT (124M) models on openwebtext-100k with batch sizes $\text{BS} = \\{8, 16, 32, 48, 64\\}$ for one epoch (the number of training tokens is fixed to 46 million). For each batch size, we independently tuned the learning rate to its best-performing values ($1.0\times 10^{-2}$ for BS=8, $5.0\times 10^{-3}$ for other BS settings), ensuring a fair comparison across different settings. As shown in training loss curve in **Figure 10 (Appendix L)**, smaller batches yield noisier trajectories while larger batches produce smoother curves, yet all settings converge to nearly the same final training and validation loss (approximately 4.0).
>
> These results demonstrate that our method is highly robust to batch-size variation: the convergence behavior and final performance are reasonably good and consistent across a wide range of batch sizes. Among the configurations, $\text{BS}=16$ provides the best model performance, which is used in the main experimental settings.
>
> **Q2. The theoretical bound includes constants depending on parameter dimensions. Can the authors clarify how large these constants are in practice, and whether they affect scalability to larger models?**
>
> **A2.** One can refer to **Lemma A.3 (in Appendix A)** and **Table 4 (in Appendix F)** for the choices of the constants $C_1$ and $C_2$, where $m, n$ in Lemma A.3 correspond to $d_{\text{model}}, d_{\text{FF}}$ in Table 4. By calculation, the constants of the dominating terms in Theorem 5.3 depend on $C_2^{3/2}/C_1$, which is about $10^3$ magnitude and not exceedingly large even if the model size is large (e.g., LLaMA-2B). In addition, we clarify that this dimension dependence is only determined by the shape of each individual matrix, meaning it is affected by the width rather than the depth of the neural network. Since larger models typically scale in depth rather than width, these constants generally do not affect scalability to larger models.

---

### Official Review · Reviewer_tPhc · 2025-11-01

**Soundness:** 3
**Presentation:** 3
**Contribution:** 2
**Rating:** 4
**Confidence:** 2

**Summary:**

This paper proposes a geometry-aware optimization algorithm named LANTON (LAyer-wise Noise-adaptive learning raTe scaling with Operator Norms). Building upon the frameworks of the D-Muon and Scion geometry-aware optimization algorithms, LANTON introduces noise-adaptive layer-wise learning rate scaling, which dynamically estimates the gradient variance in the dual norm induced by the selected Linear Minimization Oracle (LMO) and uses this estimate to assign layerwise learning rates that adapt over the course of training. This adaptive scaling enables the optimizer to assign smaller learning rates to noisier layers and larger learning rates to less noisy layers, thereby improving convergence efficiency.

**Strengths:**

1.	This paper proposes a new approach that combines geometry-aware optimization with noise-adaptive layer-wise learning rates.

2.	It achieves about 1.5× faster convergence than the baseline D-Muon in large-scale model training.

3.	The paper is clearly structured and well-organized, and Algorithm 1 is easy to understand.

**Weaknesses:**

1.	This paper essentially integrates Geometry-Aware Optimization with Layer-wise Adaptive Learning Rates. The method inherits the Geometry-Aware Optimization framework from D-Muon, and, building on that, adopts the concept from LAMB (Layer-wise) to apply noise-adaptive layer-wise learning rate scaling, assigning different learning rates to different layers within the same group. The noise estimation mechanism also resembles prior “variance-adaptive learning rate” methods such as AdaNoise, RAdam, and AdaBelief. It does not introduce any innovative improvements to the geometry-aware optimization, layer-wise adaptive learning rate, or noise adaptivity components themselves.
2.	The paper does not compare LANTON with more recent Layerwise Adaptive Learning Rate optimization methods.
3.	Lines 234–236 state that“Unlike Muon and D-Muon, which use AdamW for embedding and LM head layers, we adopt a geometry-aware framework (similar to Scion) and update these weight-sharing layers with Signum (see Table 1).” However, despite utilizing the Scion framework, the experiments do not include a direct comparison with Scion.

**Questions:**

1.	For the same group, different layers still share the same norms. What if this grouping were further refined or made dynamic?
2.	Why were experiments not conducted on larger models such as LLaMA-13B? In Appendix Section H, the paper states: “Averaged over three independent runs, D-Muon requires 20 h 05 m 24 s, whereas LANTON requires 20 h 55 m 37 s, about 4% more training time than D-Muon” if experiments were performed on larger models, would the computational overhead still remain around 4 % as reported in the paper? How would it change as the number of parameters increases?
3.	Have alternative noise-estimation methods been explored? Why was this estimation strategy chosen?
4.	Why does the experimental section not include a comparison with the Scion method?

---

> ### Author Response · Authors · 2025-11-23
>
> **Q1. This paper essentially integrates Geometry-Aware Optimization with Layer-wise Adaptive Learning Rates. The method inherits the Geometry-Aware Optimization framework from D-Muon, and, building on that, adopts the concept from LAMB (Layer-wise) to apply noise-adaptive layer-wise learning rate scaling, assigning different learning rates to different layers within the same group. The noise estimation mechanism also resembles prior “variance-adaptive learning rate” methods such as AdaNoise, RAdam, and AdaBelief. It does not introduce any innovative improvements to the geometry-aware optimization, layer-wise adaptive learning rate, or noise adaptivity components themselves.**
>
> **A1.** First, as discussed in Section 4, although optimizers such as LARS [1] and LAMB [2] employ layer-wise rescaling to stabilize large-batch training, they treat all layers uniformly and do not incorporate the notion of “groups”. In contrast, our algorithm is geometry-aware, selecting norms tailored to hidden, embedding, and normalization layers, and updating them through LMOs with noise-adaptive scaling.
>
> Second, RAdam [3] made unrealistic distribution assumptions (see their Section 4) to do variance estimation and then obtain rectified adaptive learning rate while we do not. In particular, we build upon [4] rather than RAdam [3] in estimating noise magnitude. In addition, AdaBelief [5] does not incorporate variance in the algorithm design. Also, we did not find the related literature about AdaNoise, please let us know the reference.
>
> Third, compared to the geometry-aware optimizer Scion [6], our method introduces noise-adaptive layer-wise learning rates by estimating gradient variance in the dual norm induced by the chosen LMO.
>
> **Q2. The paper does not compare LANTON with more recent Layerwise Adaptive Learning Rate optimization methods.**
>
> **A2.** We included the comparison with LAMB [2] and the recent block-wise scheme BW-AdamW [7] in Section 6.3 (Comparison with Algorithms Using Layer-wise/Block-wise Learning Rates), see Figure 4(a) for more details.
>
> **Q3. Lines 234-236 state that “Unlike Muon and D-Muon, which use AdamW for embedding and LM head layers, we adopt a geometry-aware framework (similar to Scion) and update these weight-sharing layers with Signum (see Table 1).” However, despite utilizing the Scion framework, the experiments do not include a direct comparison with Scion.**
>
> **A3.** We included the comparison with Scion [6] in Sections 6.2.2 (GPT2 on Openwebtext) and 6.2.3 (LLaMA on C4 and MiniPile), see Figures 2 and 3 for more details.
>
> **Q4. For the same group, different layers still share the same norms. What if this grouping were further refined or made dynamic?**
>
> **A4.** We follow the grouping strategy used in recent works such as Scion [6] and modular duality [8]. Specifically, we predefine groups layers based on layer type and geometry
> (e.g., attention QK/VO, MLP, embedding, normalization), and assign an appropriate norm for each group. Within each group, LANTON adjusts the learning rates of individual layers according to their estimated gradient noise, which is the core mechanism enabling our noise-adaptive behavior.
>
> Exploring finer or dynamic grouping, as well as automatically selecting the most suitable norm during training, is a promising direction that we plan to investigate in future work. We thank the reviewer for this valuable suggestion.
>
> [1] Yang You, Igor Gitman, and Boris Ginsburg. Scaling sgd batch size to 32k for imagenet training. arXiv preprint arXiv:1708.03888, 6:12, 2017.
>
> [2] Yang You, Jing Li, Sashank Reddi, Jonathan Hseu, Sanjiv Kumar, Srinadh Bhojanapalli, Xiaodan Song, James Demmel, Kurt Keutzer, and Cho-Jui Hsieh. Large batch optimization for deep learning: Training bert in 76 minutes. arXiv preprint arXiv:1904.00962, 2019.
>
> [3] Liyuan Liu, Haoming Jiang, Pengcheng He, Weizhu Chen, Xiaodong Liu, Jianfeng Gao, and Jiawei Han. On the variance of the adaptive learning rate and beyond. arXiv preprint arXiv:1908.03265, 2019.
>
> [4] Xiaochuan Gong, Jie Hao, and Mingrui Liu. Adaptive algorithms with sharp convergence rates for stochastic hierarchical optimization. arXiv preprint arXiv:2509.15399, 2025.
>
> [5] Juntang Zhuang, Tommy Tang, Yifan Ding, Sekhar C Tatikonda, Nicha Dvornek, Xenophon Papademetris, and James Duncan. Adabelief optimizer: Adapting stepsizes by the belief in observed gradients. Advances in neural information processing systems, 33:18795–18806, 2020.
>
> [6] Thomas Pethick, Wanyun Xie, Kimon Antonakopoulos, Zhenyu Zhu, Antonio Silveti-Falls, and Volkan Cevher. Training deep learning models with norm-constrained lmos. arXiv preprint arXiv:2502.07529, 2025.
>
> [7] Jinbo Wang, Mingze Wang, Zhanpeng Zhou, Junchi Yan, Lei Wu, et al. The sharpness disparity principle in transformers for accelerating language model pre-training. arXiv preprint arXiv:2502.19002, 2025
>
> [8] Jeremy Bernstein and Laker Newhouse. Modular duality in deep learning. arXiv preprint arXiv:2410.21265, 2024.

---

> ### Author Response · Authors · 2025-11-23
>
> **Q5. Why were experiments not conducted on larger models such as LLaMA-13B? In Appendix Section H, the paper states: “Averaged over three independent runs, D-Muon requires 20 h 05 m 24 s, whereas LANTON requires 20 h 55 m 37 s, about 4\% more training time than D-Muon” if experiments were performed on larger models, would the computational overhead still remain around 4\% as reported in the paper? How would it change as the number of parameters increases?**
>
> **A5.** Due to limited compute resources, we were unable to run LLaMA-13B experiments. However, we performed additional evaluations on a larger 2B-parameter LLaMA model with the training budget of 10B tokens. As shown in **Table 1** and **Figure 8 in Appendix J** (Appendix H in the original submission), the relative ratio between the overhead of LANTON and D-Muon remains almost unchanged when scaling from 1.1B to 2B:
> - 1.1B model: LANTON adds $4.18\\%$ training time over D-Muon.
> - 2B model: LANTON adds $4.04\\%$ training time over D-Muon,  which is almost identical to the 1.1B case.
>
>
> This stable overhead ratio arises because the gradient-noise estimation in LANTON is implemented with a low-rank randomized SVD ($d = 100$)  [1, 2] to approximate the nuclear norm, rather than performing a full SVD on large parameter matrices. Therefore, even for bigger matrices, because we use a fixed $d=100$, the additional overhead by evaluating the norm of the gradient is almost the same.
>
> These results and analysis indicate that LANTON’s overhead is stable at $4\\%$ and does not increase with model scale, suggesting similar behavior would be expected on models such as LLaMA-13B.
>
> **Table 1: Comparison of running time.**
>
> | **Method** | **1.1B model (hours)** | **2B model (hours)** |
> |-----------|-------------------------|------------------------|
> | D-Muon    | 20.08                   | 65.15                 |
> | LANTON    | 20.92                   | 67.78                 |
>
>
> [1] Nathan Halko, Per-Gunnar Martinsson, and Joel A Tropp. Finding structure with randomness: Probabilistic algorithms for constructing approximate matrix decompositions. SIAM review, 53 (2):217–288, 2011.
>
> [2] Tae-Hyun Oh, Yasuyuki Matsushita, Yu-Wing Tai, and In So Kweon. Fast randomized singular value thresholding for nuclear norm minimization. In Proceedings of the IEEE Conference on Computer Vision and Pattern Recognition, pp. 4484–4493, 2015.
>
>
> **Q6. Have alternative noise-estimation methods been explored? Why was this estimation strategy chosen?**
>
> **A6.** We did not explore alternative noise estimation methods, as our approach is standard way to estimate stochastic gradient variance on the fly and we are not aware of other methods.
>
> We adopt this estimation strategy because, under Option II, it allows us to establish high-probability theoretical guarantees for both $H_t^{\ell}$ and $\alpha_t^{\ell}/\alpha_t^{m}$ (see Lemmas 4.4 and 4.5), thereby ensuring the convergence of Algorithm 1.
> In practice, we approximate $\tilde{G}\_t^\ell$ by the previous step gradient $G_{t-1}^{\ell}$ (Option I in line 7). This avoids doubling the batch size and keeps the total number of sampled data consistent with standard baselines, ensuring fair empirical comparisons.

---

### Author Response · Authors · 2025-11-23
**General Repsonse**

We thank all reviewers for their insightful comments. We summarize the main updates and clarifications made in the revised version (marked in blue):

- **Clarified contributions.** We explicitly acknowledge the connections to Gluon in section of Introduction and Method, and emphasize that our main novelty lies in introducing *noise-adaptive layerwise learning rates* within geometry-aware LMOs.

- **Scalability \& overhead.** New 2B-LLaMA experiments and a refined runtime analysis in Appendix J show that the relative overhead ratio between LANTON and D-Muon remains almost unchanged ($\sim 4\\%$) when scaling from 1.1B to 2B, indicating that this overhead is independent of model size.

- **Added batch-size sensitivity study.** A new experiment on batch sizes $\\{8,16,32,48,64\\}$ (**Figure 10 in Appendix L**) shows that all settings converge to nearly the same final loss, demonstrating robustness of our gradient noise estimator.

- **Added image-classification results.** On CIFAR-100 airbench in **Appendix D**, LANTON takes much fewer epochs to achieve the best training and validation accuracy among all baselines.

- **Improved theory and notation.** We corrected notation inconsistencies, clarified the product-space setup, and added explanation for assumptions and the proof sketch.


We believe these additions address the reviewers' concerns and substantially strengthen the paper.

---

### Meta-Review · Area_Chair_2q7x · 2025-12-21

**Summary:**

This paper proposes a new technique to choose the learning rate in a noise-adaptive and per-layer fashion. It relies on online estimates of the gradient variance in the dual norm to assign smaller steps to noisier layers and larger steps to cleaner ones.

The reviews recognize the merits of the submission, especially a straightforward integration with Scion/Muon-style frameworks, and empirical gains.

The main criticisms center on novelty and the discussion/comparison to prior work: this new method is viewed by some as largely combining existing geometry-aware machinery with layer-wise scaling ideas reminiscent of LAMB and variance-adaptive optimizers, with concerns about insufficient comparison to recent layer-/block-wise methods and (initially) missing or unclear comparisons to Scion/Gluon.

In their replies, the authors claim adding proper citations/discussions of prior work and adding missing baselines (e.g. LAMB and BW-AdamW). The criticism of computational overhead is argued to be small and roughly stable with scale due to randomized low-rank SVD and infrequent noise updates.

All in all, this paper is a rather borderline case. Reviewer w5Yy gave a reject based on the similarity with Pethick et al. I do agree that this work makes heavy uses of some parts of their proof (their descent lemma is a starting point of the analysis in this paper). However, the proof then goes on to handle noise-adaptive layerwise learning rates and moving-average variance estimates and it proceeds more similarly to Cutkosky & Mehta 2020. Overall, I feel the proof has some novel aspects to it. Perhaps a more important concern that caught my attention is the lack of comparison to existing baselines. Although the authors did partially address this by adding some results (e.g. Figure 4: Training/validation loss on C4 datasets), this part feels rushed. I think it would be worth comparing these baselines across several models/datasets to get a more complete picture and add a deeper discussion of the choice of hyper-parameters. The only relevant sentence I found is for BW-AdamW: "Following the original best tuned ratio...". Please also add the variance computed accross several runs to your plots.

Overall, I find the direction taken in this paper promising. However, the reviewers raised several valid concerns that, in my view, were only partially addressed in the rebuttal. For this reason, I do not recommend acceptance at this time, but I strongly encourage the authors to improve the comparison with existing baselines before resubmitting the work.

**Reviewer Concerns:**

I think the overlap with prior work in terms of the theoretical analysis is well addressed in the rebuttal. I'm more concerned about the comparison to existing baselines. The authors only added 1 experiment to compare to LAMB and BW-AdamW. These results feel rushed, which ultimately led me to reject this paper so that more solid experiments can be presented in a future revision.

**Reviewer Scores:**

The authors have done a good job addressing most of the comments. As noted above, however, the empirical component of the paper still requires further work.

---

### Decision · Program_Chairs · 2026-01-26

Reject